

# Definitions and methods to estimate regional land carbon fluxes for the second phase of the REgional Carbon Cycle Assessment and Processes Project (RECCAP-2)

Philippe Ciais[1], Ana Bastos[2], Frédéric Chevallier[1], Ronny Lauerwald[1,3], Ben Poulter[4], Pep Canadell[5], Gustaf Hugelius[6,7], Robert B. Jackson[8], Atul Jain[9], Matthew Jones[10], Masayuki Kondo[11], Ingrid T. Luijkx[12], Prabir K Patra[13], Wouter Peters[12,14], Julia Pongratz[15], A. M. Roxana Petrescu[16], Shilong Piao[17,18], Chunjing Qiu[1], C. Von Randow[19], Pierre Regnier[3], Marielle Saunois[1], Robert Scholes[20], A. Shvidenko[21,22], Hanqin Tian[23], Hui Yang[1], Xuhui Wang[17] and Bo Zheng[1]

[1] Laboratoire des Sciences du Climat et de l'Environnement, CEA-CNRS-UVSQ-U.P.Saclay, Gif sur Yvette, France

[2] Max-Planck-Institut für Biogeochemie, Hans-Knöll-Str. 10, 07745, Jena, Germany

[3] Université Libre de Bruxelles, Department Geoscience, Environment & Society, Bruxelles, Belgium

[4] NASA Goddard Space Flight Center, Biospheric Sciences Lab., Greenbelt, USA

[5] Global Carbon Project, CSIRO Oceans and Atmosphere, GPO Box 1700, Canberra, Australia

[6] Department of Physical Geography, Stockholm University, Sweden

[7] Bolin Centre for Climate Research, Stockholm University, Sweden

[8] Department of Earth System Science, Woods Institute for the Environment, and Precourt Institute for Energy, Stanford University, USA

[9] Department of Atmospheric Sciences, University of Illinois, Urbana, IL 61821, USA

[10] Tyndall Centre for Climate Change Research, School of Environmental Sciences, University of East Anglia, Norwich Research Park, Norwich NR4 7TJ, UK

[11] Center for Global Environmental Research, National Institute for Environmental Studies, Tsukuba, Japan

[12] Meteorology and Air Quality, Wageningen University, Wageningen, the Netherlands

[13] Japan Agency for Marine-Earth Science and Technology (JAMSTEC), Yokohama, Japan

[14] Centre for Isotope Research, University of Groningen, Groningen, the Netherlands

[15] Department für Geographie, Ludwig-Maximilians-Universität München, Luisenstr. 37, München, Germany

[16] Department of Earth Sciences, Vrije Universiteit Amsterdam, Amsterdam, the Netherlands

[17] Sino-French Institute for Earth System Science, College of Urban and Environmental Sciences, Peking University, Beijing, China

[18] Institute of Tibetan Plateau Research, Chinese Academy of Sciences, Beijing 100085, China

[19] Earth System Science Center, National Institute for Space Research, Brazil

[20] Global Change Institute, University of the Witwatersrand, Johannesburg, South Africa

[21] International Institute for Applied Systems Analysis, A-2361 Laxenburg Austria

[22] Center of Productivity of Forests Russian Academy of Sciences, Moscow, Russia

[23] International Center for Climate and Global Change Research, School of Forestry and Wildlife Sciences, Auburn University, Auburn, USA

*Correspondence to*: Philippe Ciais (philippe.ciais@lsce.ipsl.fr)



**Abstract.**

Regional land carbon budgets provide insights on the spatial distribution of the land uptake of atmospheric carbon dioxide, and can be used to evaluate carbon cycle models and to define baselines for land-based additional mitigation efforts. The scientific community has been involved in providing observation-based estimates of regional carbon budgets either by downscaling atmospheric CO2 observations into surface fluxes with atmospheric inversions, by using inventories of carbon stock changes in terrestrial ecosystems, by upscaling local field observations such as flux towers with gridded climate and remote sensing fields or by integrating data-driven or process-oriented terrestrial carbon cycle models. The first coordinated attempt to collect regional carbon budgets for nine regions covering the entire globe in the RECCAP-1 project has delivered estimates for the decade 2000-2009, but these budgets were not comparable between regions, due to different definitions and component fluxes reported or omitted. The recent recognition of lateral fluxes of carbon by human activities and rivers, that connect CO2 uptake in one area with its release in another also requires better definition and protocols to reach harmonized regional budgets that can be summed up to the globe and compared with the atmospheric CO2 growth rate and inversion results. In this study, for the international initiative RECCAP-2 coordinated by the Global Carbon Project, which aims as an update of regional carbon budgets over the last two decades based on observations, for 10 regions covering the globe, with a better harmonization that the precursor project, we provide recommendations for using atmospheric inversions results to match bottom-up carbon accounting and models, and we define the different component fluxes of the net land atmosphere carbon exchange that should be reported by each research group in charge of each region. Special attention is given to lateral fluxes, inland water fluxes and land use fluxes.

**Introduction**

The objective of this paper is to define the land-atmosphere $CO_2$ or total carbon (C) fluxes to be used in the REgional Carbon Cycle Assessment and Processes-2 (RECCAP2) project. Accurate and consistent observation-based estimates terrestrial carbon budgets at regional scales are needed to understand the global land carbon sink, to evaluate land carbon models used for carbon budget assessments and future climate projections, and to define baselines for land-based mitigation efforts. In the previous synthesis called RECCAP1, regional data from inventories were compared with global models output from atmospheric inversions, process-based land models, the results being synthesized in a special issue (https://bg.copernicus.org/articles/special_issue107.html) for 9 land regions in the period 2000-2009. The definition of fluxes was not harmonized and inland waters and trade induced $CO_2$ fluxes were not considered for most regions. The RECCAP1 synthesis spurred efforts to provide new global analysis of inland water $CO_2$ fluxes (Raymond et al. 2013). Recently, Ciais et al. (2020) collected bottom up inventory estimates from RECCAP1 papers and completed them with other components, to derive the first global bottom up estimate of the net land atmosphere C exchange, that compared well with the independent top down estimate obtained from the $CO_2$ growth rate minus fossil fuel emissions and ocean uptake.



The aims of RECCAP2 are to collect and synthesize regional $CO_2$, $CH_4$ and $N_2O$ budgets for 10 continental-scale regions
(including one 'cross cutting' region consisting of all permafrost covered boreal areas), together covering the globe (Fig. 1).
There is thus a requirement for harmonization and consistency, sufficient to be able to scale regional budgets to the globe and
to compare different regions with each other for all component fluxes and each greenhouse gas. There is further an intention
to compare the results of top-down atmospheric inversions with bottom-up accounting approaches. Since research groups
working on the synthesis of greenhouse gas budgets in different regions or using different approaches use different datasets
and definitions, it is important to provide a set of shared and agreed definitions that are as precise as possible for each flux to
be reported. We focus here on land C and $CO_2$ budgets, defined from two approaches: 'top-down' estimates from atmospheric
inversions, and; 'bottom-up' carbon accounting approaches based on C stock inventories, process- and data-oriented models.

Atmospheric inversions analysis of land-atmosphere $CO_2$ fluxes inherently differs from bottom-up C budgets for two reasons.
The first one is the existence of lateral fluxes at the land surface and from the land to the ocean, which displace carbon initially
fixed as $CO_2$ from the atmosphere in one region and release it outside that region. Consequently, the $CO_2$ flux diagnosed by
an inversion is not equal to the change of stock in a region. The second one is that carbon enters from the atmosphere in the
land reservoirs almost uniquely as $CO_2$ fixed by photosynthesis, while it is released both as $CO_2$ and as reduced carbon
compounds encompassing CO, $CH_4$ and biogenic volatile organic compounds (BVOCs). Again, this process makes $CO_2$ fluxes
different from total carbon fluxes across the land-atmosphere surface.

To address these issues, Section 1 of this paper covers atmospheric $CO_2$ inversions and the treatment of reduced C compounds
emissions, with the goal to make inversion results comparable with total C flux estimates from bottom-up approaches. Section
2 deals with bottom up estimates and provides definitions of the main component land-atmosphere C fluxes that should be
estimated individually to provide a full assessment of the C balance of each region, to enable consistent comparisons between
regions and upscaling of regional budgets to the globe. Section 3 provides a description of different approaches used to derive
regional component C fluxes in different bottom-up approaches, outlining which fluxes are included or ignored by each
different approach. Section 4 gives recommendations regarding the estimation of carbon emissions resulting from land use
change, with systematic errors and omission errors associated to different approaches. We conclude by providing
recommendations for a multiple-tier approach to develop regional C budgets in RECCAP2.

**1 Top-down land-atmosphere C fluxes from atmospheric inversions**

**1.1 Land $CO_2$ fluxes covered by inversions**

The approaches known as top-down atmospheric inversions estimate the net $CO_2$ flux exchanged between the surface and the
atmosphere by using atmospheric transport models and $CO_2$ mole fraction measurements at various locations. The mole



fraction data comes from surface stations, which have been available in increasing numbers since 1957. More recently, total column mole fraction of $CO_2$ have been observed with global coverage by satellites, GOSAT since 2009 and OCO-2 since 2014 (Liu et al., 2020). Because the sampling of the atmosphere is sparse, even with the recent global satellite observations, there is an infinite number of flux combinations that can fit atmospheric $CO_2$ observations within their errors. Most inversions

therefore use a Bayesian statistical approach where an optimal $CO_2$ flux is found as a maximum likelihood estimate in the statistical distribution of possible fluxes, given a prior value and its uncertainty distribution, and observations, which also have an uncertainty distribution. The effect of fossil fuel and cement production $CO_2$ emissions (hereafter collectively called "fossil fuel" for simplicity) on mixing ratios gradients is accounted for by prescribing transport models with an assumed fixed map of fossil $CO_2$ emissions. The signal from these emissions in the space of concentrations is removed at pre- or post-processing

stage from inversions to solve for residual non-fossil $CO_2$ fluxes. Over land, output fluxes from inversions are thus the sum of all non-fossil $CO_2$ fluxes. This includes gross primary production $CO_2$ uptake, plant and soil respiration, litter photo-oxidation, biomass-burning emissions both from wildfires and for the purposes of energy provision, inland-water fluxes, the oxidative release of $CO_2$ from biomass consumed by animals and humans and decaying in waste pools, $CO_2$ emitted by insect grazing, geological $CO_2$ emissions from volcanoes and seepage from below-ground sources, $CO_2$ uptake from weathering reactions and

geological $CO_2$ release from microbial oxidation of petrogenic carbon (Hemingway et al., 2018). Inversions have very limited capability to separate those different fluxes unless they use additional information, which is not the case for inversions used in global budgets. An example of additional information is the use of CO as a tracer, to separate emissions from vegetation fires from those from fossil fuels and respiration.

### 1.2 Prescribing fossil $CO_2$ emission fields, inclusive of bunker fuels

Within RECCAP1 (Canadell et al., 2015), the same fossil fuel emission estimate was subtracted from the total posterior fluxes of participating inversions, even when those inversions had used different fossil fuel inventories (Peylin et al., 2013). This inconsistency between the inversion process and the inversion post-processing induced artifacts (see discussion in Thompson et al., 2016) but is of lesser importance for the inter-comparison than the use of different fossil fuel inventories within the inversion ensemble. We thus recommend here that a standard gridded a priori fossil fuel $CO_2$ emission estimate is used by all

regions in RECCAP2, such as recently prepared by Jones et al. (2020). Another important issue is that about 10% of $CO_2$ emissions come from mobile sources from ships at the ocean surface, and aircraft in the volume of the atmosphere. We recommend that these 'bunker fuel' emissions are prescribed to RECCAP2 inversions in using three-dimensional maps of fossil fuel $CO_2$ emissions. Each grid box should thus include the emissions within its borders, along ship routes on the surface, and flight paths at the appropriate altitude in the atmosphere. This option is increasingly viable due to the emerging availability

of sectoral emissions grids for recent years (Choulga et al., 2020; Jones et al., 2020).



### 1.3 Reduced-C compounds emissions

Reduced-C compounds are emitted by the land surface as biogenic and anthropogenic $CH_4$, BVOCs and CO. Globally, emissions of reduced C compounds from land ecosystems and fossil fuel use are a large and overlooked component of the C budget, with CO-carbon emissions from incomplete fuel combustion equaling $\approx 0.3$ PgC $y^{-1}$ (Zheng et al., 2019), $CH_4$-carbon

emissions 0.43 PgC $y^{-1}$ (Saunois et al., 2020) and non-methane biogenic compounds emissions up to 0.75 PgC $y^{-1}$ (Sindelarova et al., 2014). Given that inversions only assimilate atmospheric observations of $CO_2$, they omit regional emissions of reduced C compounds. However, reduced C compounds all oxidize to $CO_2$ in the atmosphere, with lifetimes of hours to days for BVOCs, months for CO and nearly ten years for $CH_4$. The global $CO_2$ growth rate thus includes the signal of the global reduced C emissions being oxidized into $CO_2$ in the volume of the atmosphere, though not necessarily in the year of their emission. By

fitting the global $CO_2$ growth rate, inversions thus include global emission of reduced C compounds, which is diagnosed as a diffuse natural $CO_2$ emission over the whole surface of the globe, in that year. This implies that inversions place a wrong ocean $CO_2$ emission in the place of reduced C compounds emitted only over land (Enting and Mansbridge, 1991). Further, current inversions assume that all the fossil C is emitted as $CO_2$ ignoring incomplete fuel combustion emitted as CO. The signal from fossil fuel CO emissions on the $CO_2$ concentration field is therefore incorrectly treated as a surface emission of fossil $CO_2$.

Such an overestimation of fossil $CO_2$ emissions at the surface, mainly over northern hemisphere large fossil fuel emitting regions, leads to an overestimation of the surface $CO_2$ sink in order to match the interhemispheric $CO_2$ gradient.

A mathematical formulation of the effect of CO emissions and oxidation on the latitudinal gradient of atmospheric $CO_2$, and its impact on natural $CO_2$ fluxes in a 2D inversion ignoring incomplete fuel combustion emitted as CO, which amount to $\approx 0.3$

PgC (latitude-vertical) was given by Enting and Mansbridge (1991). They showed that an inversion that includes an atmospheric CO loop of the carbon cycle placed a larger surface $CO_2$ sink in the northern tropics and a smaller surface $CO_2$ sink north of 50°N, compared to an inversion without this process. Using a 3D inversion, Suntharalingam et al. (2005) confirmed the impact of CO oxidation in the atmosphere, although with modest effects on diagnosed land $CO_2$ fluxes. We describe below an approach to correct for the effect of BVOCs, CO and $CH_4$ in inversions for RECCAP2. This approach allows

the translation of current inversions $CO_2$ fluxes into total C fluxes that can then be consistently compared with total-C fluxes given by bottom up approaches.

### 1.4 Correcting net $CO_2$ ecosystem exchange from inversions for reduced compounds

Separate corrections to inversions should be made for BVOCs, CO and CH4 because they have very different lifetimes, thus affecting in different ways the $CO_2$ mole fraction gradients measured by surface networks or satellites. Most BVOCs have a

short lifetime and are oxidized to $CO_2$ in the boundary layer. This means that inversions using $CO_2$ concentration observations interpret BVOC emissions as local surface $CO_2$ emissions. Globally, carbon emissions from VOCs amount to 0.8 PgC $y^{-1}$, mostly biogenic (Guenther et al., 2012) and dominated by isoprene, methanol and terpenes (Folberth et al., 2005). If the purpose





is to compare inversions to Net Ecosystem Exchange (NEE) of total C derived from bottom-up methods (see Section 2) we recommend to include BVOC carbon emissions in bottom-up regional estimates of NEE, rather than making BVOC correction

of inversion $CO_2$ fluxes.

Regarding the effect of the fossil CO loop of the atmospheric $CO_2$ cycle mentioned above, we propose to treat fossil CO as a 'bunker fuel'. First, we have to reduce the prescribed prior gridded fossil $CO_2$ emissions by the gridded amount emitted as CO, using space time distribution of this CO source from inventories or from fossil CO emissions inversion results. Then, we have

to prescribe a compensatory prior 3D atmospheric $CO_2$ source originating from fossil CO oxidized by OH in the atmosphere. Knowledge of thus prior 3D source of $CO_2$ from fossil origin is now available from atmospheric chemistry models used by global fossil CO emissions inversions since 2000 (Zheng et al. 2019). Other chemistry-transport models simulating the atmospheric oxidation chain of reduced C compounds unconstrained by observations may not be accurate enough for that purpose (Stein et al., 2014). We thus recommend to develop for RECCAP2 new fossil $CO_2$ emission prior field which include

the fossil CO loop. The impact of such new priors will be to reduce inversion estimates of natural $CO_2$ sinks in the northern hemisphere over regions where fossil fuels are burned, and to enhance sinks in the tropics and subtropics where CO is oxidized into $CO_2$.

Regarding the effect of CO emissions from wildfires which ranges globally from 0.15 to 0.3 PgC $y^{-1}$ (Zheng et al., 2019; van

der Werf et al., 2017), the action to be taken for inversions depends on the configuration of each system, since inversions do not all use a prior fire emission map, in which case CO from fires could be treated like CO from fossil fuels as explained above. Looking into the three global inversions used in previous global carbon budget assessments, the Jena-Carbo Scope inversion (Rödenbeck et al., 2003) does not have biomass burning a priori $CO_2$ emissions, the CarbonTracker Europe (CTE) inversion (Peters et al., 2010, Luijkx et al., 2017) prescribes temporal and spatial prior fire emissions which means that any $CO_2$ uptake

by vegetation regrowth after fire will be spread as a diffuse $CO_2$ sink within and outside burned regions and the CAMS inversion (Chevallier, 2019) prescribes temporal and spatial prior fire emissions and an annual $CO_2$ uptake equal to annual emissions over each grid cell affected by fires. This setting of CAMS forces an annual regrowth of forests after burning, yet allows the inversion to temporally allocate this regrowth uptake. CTE and CAMS consider that all prior fire emissions are $CO_2$ emissions, ignoring incomplete combustion emissions of CO. Thus, just as in fossil $CO_2$ emissions, CTE and CAMS inversions

will over-estimate the prior values of $CO_2$ mixing ratios over burned areas during the fire season. Given the lifetime of CO and given the fact that most biomass burning takes place in the tropics, prescribing all prior fire emissions as $CO_2$ in CTE and CAMS will cause only a small positive bias in prior $CO_2$ mixing ratio at tropical stations. The situation may be different for satellite inversions assimilating column $CO_2$ data. These inversions do sample $CO_2$ plumes resulting from biomass burning, but not co-emitted CO. In that case, it is expected that inversions based on satellite observations will capture biomass-burning

$CO_2$ emissions, but underestimate fire C emissions by the amount of CO emitted by fires. Carbon emitted as CO by fires will contribute after its oxidation to the global $CO_2$ growth rate. This signal will thus be wrongly interpreted by inversions as a





diffuse $CO_2$ source spread uniformly over land and ocean. For RECCAP2, we recommend to pursue research to include $CO_2$ fluxes from the fire CO-loop as a prior field, to be tested by the inversions which already have fire prior emissions in their settings.


Regarding the effect $CH_4$–carbon emitted over land and oxidized into $CO_2$ with a lifetime of 9.6 years, thus impacting the interpretation of inversion results, we separate conceptually the effects of fossil versus biogenic $CH_4$ emissions. Fossil $CH_4$ fugitive anthropogenic emissions from oil, coal and gas contribute after atmospheric oxidation to the $CO_2$ growth rate by 0.08 PgC $y^{-1}$ (Saunois et al., 2020; their top-down estimate) some years after the emission has occurred. This signal is interpreted

by inversions as a uniform surface natural $CO_2$ source over land and ocean. We thus recommend to remove that source uniformly distributed over each grid cell and each moth from inversion posterior gridded fluxes to obtain gridded natural land and ocean $CO_2$ fluxes. A more complex treatment of this fossil $CH_4$ loop of the atmospheric $CO_2$ cycle, like proposed above for the fossil CO loop is not a priority in RECCAP2 because of the small magnitude of fossil $CH_4$–carbon compared to fossil CO one. Biogenic $CH_4$ emissions from agriculture, inland waters, waste and wetlands amount globally to 0.3 PgC $y^{-1}$ (Saunois

et al., 2020; their top-down estimate) and get oxidized by OH to create a global $CO_2$ source of the same magnitude. This source will be included in inversions gridded fluxes as a spatially uniform emission over land and ocean. Nevertheless, unlike for fossil $CH_4$ emissions, this source is compensated by $CO_2$ sinks from photosynthesis over ecosystems releasing $CH_4$ (paddy rice areas, grazed lands and wetlands). Inversions will capture the global effect of these $CO_2$ sinks, but not their spatial patterns, given the low density of the surface network over $CH_4$ emitting areas. Thus, we will not recommend a correction of gridded

inversions $CO_2$ fluxes for the effect of biogenic $CH_4$-carbon emissions.

### 1.5 Adjustment for 'lateral fluxes' in $CO_2$ inversions to compare them with bottom-up C budgets

With the above-recommended treatment of reduced-C emissions, inversions in RECCAP2 will provide gridded and regional means of land atmosphere C fluxes. Inversions form a complete approach, but to compare their regional C fluxes with bottom C stock changes, attention needs to paid to lateral C fluxes, as done partially by Kondo et al. (2020) and Piao et al. (2018) and

comprehensively by Ciais et al. (2020) for RECCAP1 regions. For conversion of C storage change to land-atmosphere C fluxes using lateral fluxes, we recommend to use the same methodology than in Ciais et al. (2020). The section below defines bottom-up C budgets in a way that makes it possible to match them with inversion results.

### 2 Bottom-up carbon budgets

Bottom-up approaches encompass various methods to quantify regional C budgets and their component fluxes. There is no

single observation-based bottom-up method giving comprehensively all terrestrial $CO_2$ or C fluxes. The currently- incomplete scope of existing bottom-up estimates is a source of uncertainty when trying to combine top-down with bottom-up, or when using one of these approaches to verify the results of the other (Kondo et al. 2020; Ciais et al. 2020). For improving the





completeness of regional bottom-up C budgets in RECCAP2, we define below a reasonable number of component C fluxes that can all be estimated from observations. In most cases, full observation-based estimates of component C fluxes are not feasible, but limited observations can be generally extrapolated using empirical models to the scale of RECCAP2 regions.

Figure 2 displays the required set of component C fluxes between the land and the atmosphere to be estimated for each region. No unique dataset or method is imposed to estimate each individual C flux, but we give wherever possible references of existing datasets that already quantified those fluxes. Two criteria informed the selection of C fluxes that we recommend for reporting in the RECCAP2 budgets: 1) there exists at least one estimate of each flux available at regional scale that can be used as a default Tier in the case where no regional new estimate can be obtained; 2) each flux is a non-negligible component of the global land C budget, typically an annual flux larger than 0.1 Pg C yr$^{-1}$ and thus cannot be ignored. If more detailed C fluxes are available for some RECCAP2 regions, we recommend these to be regrouped into the categories shown in Fig. 2, and this grouping to be described.

The general recommendation is, where possible, to provide several estimates for each C flux, based on different approaches. This could take the form of ensemble medians and ranges from different models. In the case where one estimate is thought to be more realistic than others, for instance a model with a better score when benchmarked against observations or a higher spatial resolution dataset with better ground validation, the underlying reasons for preferring that estimate need to be explained, based on peer reviewed literature or evaluation. Uncertainty can be calculated from the spread of different estimates, in those cases where the state of knowledge cannot establish that one estimate is better than another. The use of IPCC methods (Mastrandrea et al., 2011) and uncertainty language (http://climate.envsci.rutgers.edu/climdyn2013/IPCC/IPCC_WGI12-IPCCUncertaintyLanguage.pdf) is recommended when different estimates of the same component C flux are available. If different estimates report their own uncertainty, either based on data or an evaluation of the method used, e.g. by performing sensitivity analysis through changing model parameters, input datasets, randomly varying input data, this information should be used to evaluate consistency between estimates, given their uncertainties. It is recommended to use the word 'uncertainty' when comparing different estimates and 'error' for the difference between an estimate and true values. Because 'truth' is unknown for component C fluxes at the scale of large regions, errors cannot be estimated in RECCAP2.

## 2.1 Net carbon stock change

The net carbon stock change of terrestrial ecosystems C pools in a region (ΔC in Fig. 2) can be obtained by repeated inventories of live biomass, litter (including dead biomass), soil carbon and of carbon stock change in wood and crop products. None of the RECCAP2 region has a complete gridded inventory of all carbon stocks and their change over time. Some regions, like North America, China, Europe, Russia have forest biomass inventories established long ago by forest resource agencies (Goodale et al., 2002; Pan et al., 2011). A few countries e.g. England and Wales (Bellamy et al., 2005) and France (Martin et al., 2011) have repeated soil C inventories that allow trends to be quantified. May other countries have single-time soil carbon





inventories (e.g. US, Australia, Germany). Many regions are able to make estimates of carbon stocks in products, from forestry, wood use and crop production statistics.

For RECCAP2, we recommend that each region reports carbon stock changes in all the listed terrestrial ecosystem aggregated pools in Fig. 2, namely $\Delta C_{forest}$, $\Delta C_{croplands}$, $\Delta C_{grasslands}$ and $\Delta C_{others}$, and specify which sub-pools are include in each case. The sub-pools can include, but are not limited to the following: biomass, litter and woody debris, and soil mineral and organic carbon. Where attribution of these pools or sub-pools to biomes, land cover types, or political units is made by a regional synthesis group, the corresponding areas involved must be systematically reported. This includes the definition of the reporting depth for soil C stocks (0-30 cm and 0-100 cm are recommended). The choice of how many biomes are reported needs to

balance data availability with the importance of carbon stock and carbon stock changes within particular biomes (typically a reported biome should contribute at least 10% of the regional C changes). Regions with significant wetland C or permafrost C stocks may report this C stock separately, especially in the case where the areas involved occur in different biomes, but this must be done in a way that allows the C stocks to be subtracted from the biome total, or added back into it, without double counting. The area of biomes for which no carbon storage or carbon storage change is available needs to be reported and a

default value of -9999 should be given to such stocks and their stock change value. The biomes with no data can be specified (preferable if the area and stock involved is potentially large, since this identifies gaps needing future work), or simply lumped under 'others' if they are minor.

The net C stock change of biological products pools also needs to be reported for crop, wood and other carbon-containing

products (see Fig. 2). The depletion of peat C stocks for use as a fuel $\Delta C_{peat\ use}$ in Fig. 2 and thus causing C emissions to the atmosphere, was significant in the early 20th century in some northern countries, and still is today in few countries (Conchedda and Tubiello, 2020). It should be reported where relevant, using regional data if available (Joosten, 2009). In the case of C stock change in wood products ($\Delta C_{wood\ products}$), if possible the change in those wood products in use (e.g., construction, paper) should be reported separately from those in waste, undergoing decay (e.g. landfills). The names and definitions of the wood

product pools considered should be specified. The C stock change of crop product pools ($\Delta C_{crop\ products}$) on an annual time scale is usually small. It can be reported if data are available, otherwise a value of zero can be assumed. The net carbon stock change as organic carbon accumulation in lakes and reservoirs, known as burial. ($\Delta C_{burial}$) should be reported based on regional data or global estimates (Mendonça et al., 2017, Maavara et al., 2017).

## 2.2 Lateral displacement fluxes within and between regions

One of the reasons why net land-atmosphere C exchange excluding fossil fuel emissions, hereafter called Net Ecosystem Exchange (NEE) of a region is not equal to the net carbon stock change in the same region is because of lateral C fluxes, as alluded to in Section 1.5. Carbon is lost by each region to the adjacent estuaries through river export; lost or gained through the trade of crop, wood and animal products; and through the atmospheric transport and deposition of C particles emitted with





dust in dry regions. In order to allow the net C stock change estimates to be corrected, we recommend that lateral fluxes in and

out of each RECCAP2 region be reported. The main ones are river C export and those from wood and crop trade, as denoted by the red arrows in Fig. 2. A strong point of the RECCAP2 project is an attempt at mass balance closure between pools and fluxes. Therefore, lateral displacement fluxes of C *within* each region, but between pools denoted by the brown arrows in Fig. 2, should also be reported or calculated by mass balance. More details on these fluxes is given below.

### 2.2.1 Riverine carbon export to estuaries and the coastal ocean

Lateral C export fluxes in rivers ($F_{rivers}$ in Fig. 2) should be reported at the interface between rivers and estuaries. We recommend to top the 'land' at the mouth of rivers, and to take estuaries being coupled to the coastal ocean by dynamical and biogeochemical processes as 'blue carbon' in RECCAP2. Mangroves and salt marshes export large fluxes of dissolved and particulate C produced in upland systems or within riverine systems to estuaries and the coastal ocean (Bauer et al., 2013). These fluxes determine the carbon budget of the aquatic coastal margin ecosystems and we recommend that they should also

be considered as 'blue carbon'. River C fluxes at the river mouth into estuaries can be estimated from dissolved organic carbon (DOC), dissolved inorganic carbon (DIC) and particulate organic carbon (POC) concentration data for the rivers involved, and the associated river flow rates (Ludwig et al., 1998; Mayorga et al., 2010; Dai et al., 2012). Few RECCAP2 regions (Fig. 1) receive C by rivers entering their territory. If this is the case, this input of flux of fluvial carbon from rivers should be reported, even though for simplicity it is not represented in Fig. 2. Evasion from aquatic systems to the atmosphere is treated in Section

315 2.2.7.

### 2.2.2 Inputs of carbon to riverine from soils and weathered rocks

The inland water carbon cycle receives C leached or eroded from soils as an input. This carbon can be redeposited and buried in the freshwater ecosystems, outgassed to the atmosphere or exported to estuaries and the coastal ocean. This flux is called $F_{bio\ river\ input}$ in Fig. 2. It cannot be measured directly at large spatial scales. We therefore recommend to calculate it by mass

balance as the sum of burial, outgassing and export. Similarly, weathering processes consume atmospheric $CO_2$ (see Section 2.7). This C is subsequently delivered as dissolved bicarbonate ions to rivers. At the global scale and over long timescales, the average proportion of bicarbonate in waters is two-thirds derived from atmospheric C and one third from lithogenic C. We recommend to calculate this weathering-related DIC flux called $F_{litho\ river\ input}$ in Fig. 2, using geological maps and global weathering rates (Hartmann et al., 2009).

### 325 2.2.3 Carbon fluxes in and out each region due to trade

Net trade related C fluxes for wood and crop products exchanged by each region with others need to be reported in C units, using statistical economic data on trade volume and the carbon content of each product. These are available from regional datasets or using FAOSTAT and GTAP data, or the global dataset of (Peters et al., 2012). This net trade flux should be reported separately for crop products and wood products ($F_{crop\ trade}$ and $F_{wood\ trade}$ in Fig. 2). If relevant it can be reported for animal





products as well - but this flux is much smaller than that in crops and wood, and is therefore and is not shown in Fig. 2. Our best-practice recommendation is to separate the net trade C flux into gross fluxes of imports and exports. The list of commodities included and ignored should be specified where they are material; commodities making a small contribution can be lumped under 'other'.

**2.2.4 Crop and wood product transfers within in each region**

Figure 2 links the C stock change of terrestrial ecosystem pools to the change of C storage in biological wood products by the harvest and lateral displacement of crop and wood. The harvest of grass for forage can be assumed to be given to animals locally and can be included in $F_{grazing}$ (see details in Section 2.4). We recommend reporting the total amount of C harvested as wood and crops in each region as $F_{wood\ harvest}$ and $F_{crop\ harvest}$ (Fig. 2). Subtracting trade fluxes from the harvest fluxes will provide the C flux displaced within each region for domestic activities. Note that non-harvested and non-burned residues for

crops and forests harvesting, such as slash and felling losses should not be part of the harvest flux and should rather be counted as part of $F_{LUC}$ and $F_{land\ management}$. We note that this locally-decomposing flux is globally large, in 2000 amounting to 1.5 Pg C y$^{-1}$ for crop residues and 0.7 Pg C y$^{-1}$ for felling losses in forests (Krausmann et al., 2013).

**2.3 Net Ecosystem Exchange**

More than a decade ago, there were a number of papers trying to reconcile different definitions of land carbon fluxes (NEE,

NEP, NBP, NECB, etc.). Particularly, the papers by Schulze et al. (2000), Randerson et al. (2002), and Chapin et al. (2006). Schulze et al. focused on the importance to account for disturbance C losses at site scale when considering an ecosystem over a long time period, hence to separate Net Ecosystem Production (NEP = Gross Primary Productivity minus Ecosystem Respiration) from Net Biome Production (NBP or Net biome productivity = NEP minus disturbance emissions). Randerson et al. argued that the net carbon balance should be described by a single name NEP, provided that this flux includes all carbon

gains and losses at the spatial scale considered. Last, Chapin et al. in a 'reconciliation' paper proposed to use Net Ecosystem Carbon Balance (NECB) for the net C balance of ecosystems at any given spatial or temporal scale, and to restrict the use of NEP to the difference between Gross Primary Productivity minus Ecosystem Respiration. Those three definitions consider the C balance from the point of view of ecosystems. Here we seek to estimate the atmospheric C balance of ecosystems, at the spatial scale of large regions and the temporal scale of one decade which we call Net Ecosystem Exchange (NEE). NEE is

defined as the exchange of all C atoms between a land region and the atmosphere over it, excluding fossil fuels and cement production emissions. We use a similar definition than Hayes (2012), extended to include natural geological emissions and sinks, acknowledging that geological fluxes are not from ecosystems per se. NEE includes biogenic atmospheric emissions of CO, $CH_4$ and VOCs, all expressed in C units. This definition of NEE matches the land-atmosphere flux of total C that inversions estimate, provided they account for $CO_2$, $CH_4$, CO and VOC fluxes. NEE cannot be derived using the bottom-up approach

from a single observation-based approach. Various bottom-up datasets and methods must be combined to obtain each component flux, then those fluxes can be summed up to NEE.




We acknowledge that the geological fluxes are not strictly speaking from ecosystems, and we could therefore have called this flux net terrestrial carbon exchange rather than NEE, but the former terminology could be ambiguous, since some might assume that it includes fossil fuel and cement. NEE also includes biogenic emissions of CO, $CH_4$ and VOCs, all expressed in C units. This definition of NEE matches the land-atmosphere flux of total C that inversions estimate, provided they account for $CO_2$,

$CH_4$, CO and VOC fluxes. NEE cannot be derived using the bottom-up approach from a single observation-based approach. Various bottom-up datasets and methods must be combined to obtain each component flux, then those fluxes can be summed up to NEE.

We recommend that when a component C flux of NEE contains meaningful amounts of C emitted as CO, $CH_4$ and VOC, the

type and fraction of reduced carbon compound emitted should be reported. For instance, $F_{grazing}$ emits carbon partly as $CH_4$, $F_{fires}$ emits CO (and a smaller component of $CH_4$), VOCs and $CH_4$, $F_{wood\ products}$ emits CO when burned and $CH_4$ when the products decay in landfills (see Section 2.5), $F_{rivers\ outgas}$, $F_{lakes\ outgas}$ and $F_{estuaries\ outgas}$ emit $CH_4$ (see Section 2.6) and $F_{geological\ emissions}$ emitting $CH_4$ as well as $CO_2$ (see Section 2.7). The $CO_2$ and reduced C composition of each flux should be reported separately for clarity, both expressed in C units. This level of detail in the reporting will allow a precise comparison with

inversion fluxes (see Section 1).

In Fig. 2, the component fluxes that sum to NEE are subdivided for four sub-systems: terrestrial ecosystems, biological products, inland waters and geological pools (excluding those mined for fossil fuel and cement production). The section below describes the C fluxes components of NEE in each sub-system.

**2.4 Component fluxes of net ecosystem exchange for terrestrial ecosystems**

**2.4.1 Net Primary Productivity**

Net primary productivity (NPP) is the flux of carbon transformed into biomass tissues after fixation by GPP. NPP can be measured in the field using biometric methods, but this method does not measure non-structural carbohydrates, and NPP-acquired carbon lost to exudates, herbivores, leaf DOC leaching, biogenic VOC emissions, and $CH_4$ emission by plants (Barba

et al., 2019). Field measurements thus estimate the biomass production (BPE = sum of carbon in leaves + wood + roots), which is lower than NPP. Different satellite products provide global maps of NPP for the past decades, but the conversion of GPP to NPP is usually made by an empirical carbon use efficiency model (ratio of GPP to NPP) like the BIOME-BGC model for the GIMMS-NPP (Smith et al., 2015) and for MODIS-NPP (Running et al., 2004) or the BETHY-DLR (Wißkirchen et al., 2013a)





global products. Field-estimates of BPE can also be combined with satellite products of GPP to derive NPP (Carvalhais et al.,
2014). Discussing uncertainties of satellite NPP and GPP products is not in the scope of this report, but light-use efficiency
formulations used in many datasets tend to ignore the effect of $CO_2$ fertilization and of soil moisture deficit, which has
motivated attempts to use data-driven models or hybrid models combining process-based leaf-scale photosynthesis models
with satellite data, e.g. FAPAR, like in the P-MODEL (Stocker et al., 2019) or the BESS model for GPP (Jiang and Ryu,
2016). Those models assimilate satellite observations but include the effects of $CO_2$, diffuse light, or water stress on
photosynthesis.

Additional methods can be used to estimate regional NPP. For crop NPP, aggregated estimates can be obtained from yield
statistics and allometric expansion factors (Wolf et al., 2011), the spatial scale being the one at which yield data can be collected
(e.g. farm, county, province, country). For forest NPP, woody NPP can be obtained from forest inventories, some of the sites
having several decades of measurements enabling studies of trends. The recommendation for RECCAP2 is to document as
precisely as possible the definition of NPP in the datasets that will be used for each region, and the ecosystems covered in case
of NPP estimates limited to specific ecosystems. Also make it explicit how NPP datasets were obtained and what their possible
limitations are. We recommend that NPP and not GPP should be reported for each region, given that C from NPP links directly
to biomass and soil C inputs, and to partial appropriation by humans and animals in managed ecosystems, harvested C being
further displaced laterally and turned into emissions of C to the atmosphere where it is used.

### 2.4.2 Carbon emissions from soil heterotrophic respiration R

Soil heterotrophic respiration (SHR) is the C emitted by decomposers in soils and released to the atmosphere. Up to recently,
this flux could not be estimated directly but the availability of point scale measurements from 6000 sites (total soil respiration)
and ≈500 for heterotrophic respiration many peer-reviewed literature in the SRDB 4.0 database (Bond-Lamberty, 2018) allows
regional and global up-scaling of this flux for averages over a given period (Hashimoto et al., 2015; Konings et al., 2019;
Warner et al., 2019) or with annual variations (Yao et al. 2020) that can be used for RECCAP2.

### 2.4.3 Carbon fluxes from land use change and land management

The net land use change flux called $F_{LUC}$ includes C gross fluxes exchanged with the atmosphere from gross deforestation,
legacy and instantaneous soil $CO_2$ emissions, forest degradation emissions, and sinks from post-abandonment regrowth and
afforestation/reforestation activities (Houghton et al., 2012). This flux can be positive or negative depending on the region
considered and the balance of gross fluxes. The net land-use change flux results from changes in NPP, SHR and deforestation
fires over areas affected by land use change in the past. In absence of local NPP and SHR measurements over areas subject to
land use change, $F_{LUC}$ should be treated as a separate flux component of NEE in each region. $F_{LUC}$ is widespread in all
RECCAP2 regions and highly uncertain, and its estimates depend on the approach used. More details on the calculation of
$F_{LUC}$ are given in Section 4 since estimates of this flux depend on the method used.





The carbon flux exchanged with the atmosphere from management processes, called $F_{management}$, includes a wide range of forest, crop, and rangeland management practices. It is extremely difficult to separate $F_{management}$ from $F_{LUC}$ as it would require to quantify C fluxes from land use change followed by no management in the new land use in $F_{LUC}$, and C fluxes from additional

management activities on top of land use change. In practice, bookkeeping models of $F_{LUC}$ include management of new land use types in the empirical data they use. For instance, forest to cropland land use emissions are based on empirical observations of soil C changes in croplands from multiple sites, which implicitly include tillage, fertilization, cultivars effects but do not separate each of these practices explicitly in each region, due to lack of data. Likewise, $F_{management}$ is not simulated separately in global studies based on DGVM models, and the effects of management are included in $F_{LUC}$ instead, based on the idealized

parameterizations of management practices (Arneth et al., 2017). For croplands, DGVM models include crop harvest preventing the return of residues to soils, and some models represent tillage (Lutz et al., 2019) and changes in fertilization (Olin et al., 2015). To our knowledge, there is no DGVM simulating the effect of irrigation, changes of cultivars and rotations (cover crops), and conservation agriculture on C fluxes. For managed forests, several global models include wood harvest (Arneth et al., 2017; Yue et al., 2018) as a forcing do not have a detailed representation of practices, mainly due to the lack of

forcing data, although management is represented in some regions (Luyssaert et al., 2018). For pastures, few models include variable grazing intensity, fertilization and forage cut (Chang et al., 2015). In addition to structural DGVM limitations and lack of representation of management precluding an estimate of $F_{management}$ there is no framework to perform factorial simulations with and without land use change and management that would allow to separate $F_{management}$ and $F_{LUC}$.

$F_{LUC}$ and $F_{management}$ are accounted for by UNFCCC national communications of C fluxes in the LULUCF sector for managed lands. UNFCCC national communications report land use change emissions in their Common Reporting Format (CRF) communications for different bi-directional land-use transitions. These estimates of $F_{LUC}$ have a different system boundary from those simulated by bookkeeping models (Grassi et al., 2018; Hansis et al., 2015; Houghton and Nassikas, 2017). National communications following the IPCC guidelines (Dong et al., 2006) usually do not consider $F_{LUC}$ from land use that occurred

more than 20 years before the reporting period, whereas bookkeeping models and DGVMs consider all land use transitions that occurred since 1700. On the other hand, national communications include $F_{LUC}$ from the expansion or urban areas, which is ignored in bookkeeping models and DGVMs. In national communications, $F_{management}$ as defined here is not separately estimated. Its effect is implicitly included in the LULUCF sector based on empirical emission factors that include management practices in the new land use types, in reports of C fluxes of stable land use types (e.g. cropland remaining croplands). Since

75% of the global land ecosystems are managed (Ellis et al., 2010; Liang et al., 2016), it will be a major challenge in RECCAP2 to account comprehensively for $F_{LUC}$ and $F_{management}$ and even more so to reach a harmonized way for comparing estimates between regions. We thus recommend for each synthesis chapter to describe as precisely as possible the components of $F_{LUC}$ and $F_{management}$ and to explain in which cases they are combined together. Note that the emissions of wood products, crop





products and grazing are recommended to report as separate fluxes. If they are provided as part of $F_{LUC}$ and $F_{management}$ they
should thus be identified separately.

### 2.4.4 Carbon emissions from fires

This flux called $F_{fires}$ represents the emission of all carbon species to the atmosphere from wildfires, prescribed fires, biomass burning, and biofuel burning including $CO_2$, $CO$, $CH_4$ and black carbon, separated if possible into crop residues burning and other fires. The burning of crop residues occurs though small-scale fires, which continue to be underestimated by global
satellite burned area products. Further, some residues are burned out of the field and those emissions are not measurable with satellites. Burning emissions from crop residues can be calculated from fuel consumption and carbon emission factors. Emissions from other fires can be estimated by ground based/aerial surveys (several countries perform such surveys) or from satellite-based datasets based on burned areas such as GFED (van der Werf et al., 2010) ([www.globalfiredata.org](http://www.globalfiredata.org)), or based on fire radiative power such as GFAS (Di Giuseppe et al., 2018). GFED4.1s is an update of the GFED3 product, with updated
burned area and complemented by an active fire detection algorithm that improves detection of small fires (van der Werf et al., 2017). In tropical regions, deforestation causes fires (including peat fires in South-East Asia). It is important here to avoid double accounting by checking in each region if C emissions from deforestation fires were already included in land use change emissions $F_{LUC}$, and, if this is the case, they must be subtracted from $F_{fires}$.

### 2.4.5 Carbon emissions from insects grazing and disturbances

This flux called $F_{insects}$ represents C emissions to the atmosphere associated with background grazing and sporadic outbreak of insects. It is a significant C emission in regional budgets, though it is usually ignored, and may be estimated as a fraction of NPP or leaf biomass, if data is available, and provided no double counting, or ignored. Insect outbreaks (Kautz et al., 2017) cause direct and committed emissions to the atmosphere beyond the background grazing of a fraction of biomass, as they partly destroy foliage or cause tree morality (e.g. bark beetles in Canada, Kurz et al., 2008) that induce legacy emissions that can last
for several decades. To our knowledge, only few regions have estimates of insects-disturbances induced C emissions at regional scale, e.g. US (Williams et al., 2016), Canada, and some countries in Europe, and this component flux may not be possible to estimate for each RECCAP2 region, in particular the tropical ones.

### 2.4.6 Carbon emissions from reduced carbon species

This flux called $F_{reduced}$ is the sum of emissions to the atmosphere of reduced C compounds, including biogenic $CH_4$, biogenic
non-methane biogenic volatile organic compounds (BVOC) and biogenic CO (excluding fires). Carbon emitted as $CH_4$ by wetlands, termites, rice paddy agriculture sources and removed by soils can be estimated by bottom-up approaches, e.g. synthesized in the global $CH_4$ budget or from atmospheric $CH_4$ inversions in the case where those inversions report those flux components separately (Saunois et al., 2020). In the framework proposed here, $CH_4$ emissions from crop and wood products in landfills are counted $F_{crop\ products}$ and $F_{wood\ products}$ and $CH_4$-carbon from animals and manure in $F_{grazing}$. Emissions of carbon





from BVOC and CO by the vegetation can be obtained from models used to simulate those fluxes for atmospheric chemistry, after conversion into units of carbon mass. For instance, the CLM-MEGAN2.1 model (Guenther et al., 2012) estimates biogenic emissions of CO and of ~150 BVOC compounds with the main contributions being from terpenes, isoprene, methanol, ethanol, acetaldehyde, acetone, α-pinene, β-pinene, t-β- ocimene, limonene, ethene, and propene.

**2.4.7 Carbon emissions from biomass grazed by animals**

This flux called $F_{grazing}$ represents the C emission that incurs from the consumption of herbage by grazing animals, including the decomposition of animal products used in the bio-economy, the decomposition of manure and direct animal emissions from digestion. Only the fraction of manure from animals grazing on grass should be accounted for because C emitted from manure originating from crop-products given to animals is already included in $F_{crop\ products}$. Grass requirements by animals can be derived from grass biomass use datasets (Herrero et al., 2013). Grass biomass use per grazing animal head in a region can

be calculated based on data of total metabolizable energy (ME) of ruminants in each region. Actual grass intake can be derived from empirical models or from vegetation models that include management of pasture (Chang et al., 2016). Carbon emitted from grazed grass biomass includes $CH_4$ emissions from manure C (excreta) and from enteric fermentation, animal $CO_2$ respiration from grass intake, and C emissions from the consumption and decay of meat and milk products derived from grass grazing. The C in milk, animal and manure products can be assumed to decay in one year and to be emitted as C to the

atmosphere. Here 'animals' are domestic or wild mammals, but not insects.

**2.5 Component fluxes of net ecosystem exchange from biological products**

**2.5.1 Carbon emissions from crop biomass consumed by animals and humans**

This flux called $F_{crop\ products}$ represents the carbon emissions to the atmosphere from the consumption of harvested crop products. It can be calculated from agricultural statistics as the sum of domestically harvested products minus net export minus

storage in each region. Crop products are consumed both by animals (including wild animals) and humans, and a distinction may be made between these two groups of consumers if additional data on consumption type are available in each region. The digestion of crop products by ruminants emits $CH_4$-carbon and double counting must be avoided in case this $CH_4$-C flux is included in another C flux, like ruminant methane emissions. A fraction of C in consumed crop products is also channeled to sewage systems and lost to rivers as DOC instead of being emitted to the atmosphere, globally 0.1 PgC yr$^{-1}$ (Regnier et al.,

2013). Although it is a small flux, we recommend to include it in regional budgets if data is available. River $CO_2$ outgassing flux estimates should contain the fraction of this sewage C flux returned back to this atmosphere.

**2.5.2 Carbon emissions from harvested wood products used by humans**

This flux called $F_{wood\ products}$ represents a net carbon emission to the atmosphere from the decay and burning of harvested wood products used for paper, furniture, and construction. The emission from decay, $F_{wood\ products\ decay}$, can be calculated with models





of the fate of wood products in the economy (Eggers, (2002), Mason Earles et al., (2012)) forced by input to products pools from domestic harvest of non-fuel wood net export of wood products. The small fraction of wood-product waste going to sewage waters and rivers can also be estimated if relevant data are available. If $F_{wood\ products\ decay}$ is calculated in carbon units, e.g. from a model of wood product pools, it also includes carbon lost to the atmosphere as $CH_4$ in landfills, thus double accounting must be avoided in case $CH_4$-C emissions from wood in landfills are also reported separately in a region. The flux

from burning of wood products, $F_{wood\ product\ burning}$ can be estimated from statistics of fuel wood consumption and carbon emission factors during combustion (including $CO_2$, $CO$ and $CH_4$). This flux should include emissions from commercial fuelwood burned to produce electricity, and non-commercial fuel wood gathered locally and burned in households, and fuel wood burned as a fuel by industry. It is important to note that we recommend here to report $F_{wood\ products}$ for each RECCAP2 region as a separate flux. This term is usually included in $F_{LUC}$ in C budget studies based on DGVMs and bookkeeping models

(Friedlingstein et al., 2019). It should then be removed from currently reported estimates of $F_{LUC}$ in order to avoid double counting.

### 2.6 Component fluxes of net ecosystem exchange for inland waters

### 2.6.1 Carbon emissions from rivers, lakes and reservoirs

The fluxes called $F_{rivers\ outgas}$ and Flakes +reservoirs outgas in Fig. 2 correspond to those from the outgassing of C from lakes

and rivers, respectively. There are two global observation-based estimates of this flux calculated using the same GLORICH river $pCO_2$ database, but with different data selection criteria and up-scaling techniques. The one of Raymond et al. (2013) was produced using the COSCAT regions that represent groups of watersheds, and can be re-interpolated to the RECCAP2 regions. The one of Lauerwald et al. (2015) was produced on a $0.5° \times 0.5°$ global grid and does not include lakes. Gridded $CO_2$ emissions of boreal lakes have been estimated separately by Hastie et al. (2018) using an empirical model trained on $pCO_2$

data from mainly Swedish and Canadian lakes. The riverine $CO_2$ evasion outgassing flux from Lauerwald et al. (2015) is about half that of Raymond et al., due to lower estimates of average river $pCO_2$ for the tropics and Siberia resulting from a more restrictive data selection process and additional averaging effects from the statistical model applied. In addition, the estimates by Lauerwald et al. (2015) do not account for $CO_2$ emissions from headwater streams, which may be substantial. For instance, Horgby et al. (2019) estimated that mountain streams alone emit about 0.15 PgC $y^{-1}$ globally. Some land models have been

developed to include the land to ocean loop of the carbon cycle and their output may be used to assess river and lakes $CO_2$ evasion fluxes for selected regions (Hastie et al., 2019) or the globe. These models have also confirmed previous observational findings (e.g. Borges et al., 2015) that river floodplains are a potentially significant, yet overlooked component of the inland water C budget. Up until now, however, only $CO_2$ outgassing from rivers, lakes and reservoirs has been considered in regional C budgets. New synthesis estimates of $CH_4$ emissions from those inland waters are now available from the $CH_4$ budget

synthesis (Saunois et al., 2019) and we recommend that this source in C units should be added to $F_{rivers\ outgas}$ and $F_{lakes\ +reservoirs\ outgas}$.



### 2.7 Component fluxes of net ecosystem exchange from geological pools

### 2.7.1 Geological carbon emissions

This flux called $F_{geological\ emissions}$ correspond to natural emissions of $CO_2$ and $CH_4$ from geological pools. The Earth's degassing

of geological carbon consists of geogenic $CO_2$ emissions of 0.16 PgC $y^{-1}$ (Mörner & Etiope, 2002), microbial oxidation of rock carbon (Hemingway et al., 2018) and $CH_4$ emission estimated to be 0.027 Pg C $y^{-1}$ (Etiope et al., 2019) but recently revised (Hmiel et al., 2020) to a smaller value of 0.0012 Pg C $y^{-1}$. Geogenic $CH_4$-C land emissions are from volcanoes, mud volcanoes, geothermal sources, seeps and micro-seepage, and if the gridded dataset of Etiope et al. (2019) is used, we recommend to remove the marine coastal seepage $CH_4$-C emissions reported separately in this dataset. Geogenic $CO_2$-C emissions are almost

exclusively related to geothermal and volcanic areas (high-temperature fluid-rock interactions, crustal magma and mantle degassing). We suggest here to report these fluxes if there is a published estimate in the region considered.

### 2.7.2 Weathering uptake of atmospheric $CO_2$

This flux called $F_{weathering\ uptake}$ corresponds to the weathering of carbonate and silicate rocks which is a net sink of atmospheric $CO_2$, and corresponds to C then transferred by rivers to the ocean. We recommend that these fluxes should be reported for each

region as they are needed to rigorously compare the output of $CO_2$ inversions (which cover all $CO_2$ fluxes) with bottom-up NEE estimates (Fig. 2). This can be achieved using for instance the global dataset from Hartmann et al., (2009) and the gridded product of Lacroix et al., (2020). Weathering of cement is represented in Fig. 2 and should be reported as part of fossil fuel emissions, which is not the scope of this paper

### 3 Methods to estimate bottom up components of NEE

The methods described here are:
- C stock changes from ground based estimates (forest biomass and soil carbon inventories)
- $CO_2$ fluxes measured by Eddy-Covariance
- Other ground-based measurements (e.g. $pCO_2$ in rivers, site NPP, soil respiration data)
- Models driven by statistical data (e.g. wood and crop products and grazing emissions)
- Models driven by satellite data (e.g. fire emissions models, NPP models)
- Process-based terrestrial carbon cycle models (e.g. TRENDY models)

The general approach of RECCAP2 is to use more than one of these approaches for each flux, to gain further insights into the carbon budget of a region by exploring the full range of data available. The purpose of this section is to describe what each

method does and does not estimate in terms of NEE component C fluxes as defined in Section 2 and illustrated in Fig. 2, and therefore what valid comparisons can be made.





### 3.1 Inventory-based measurements of carbon stock changes

This approach generally uses biomass determined from repeated forest inventories. The stock changes for the LULUCF sector in UNFCCC communications reports are usually based on inventories. In some countries these have been done for many years, but in many countries, they are not available. The sampling density and sampling schemes vary greatly between countries and regions (Pan et al., 2011). The Global Forest Biomass Biodiversity Initiative (https://www.gfbinitiative.org) contains 1.2 million forest plots, mainly in the Northern Hemisphere countries, although data are currently not publicly available. The forest inventory data for tropical regions typically comes from research plots, rather than production forests. Forest inventories measure aboveground biomass, from which C stocks can be derived (and stock changes in case of repeated census) but do not quantify soil carbon changes. Repeated inventories of soil carbon only exist in very few countries or regions; where they do, they are often focused on agricultural soils alone. If site history information is available, the repeated inventories of biomass and soil C can be used to $F_{LUC}$ over time, for various land practices.

Point-scale data from inventories can be up-scaled (by simple averaging, or including spatial trends and covariates by using geo-statistics, or more recently by using machine learning) to provide regional budgets of C stock changes in biomass and soils. Forest biomass inventory estimates of tree mortality can further be used to estimate C stock changes for pools which are not directly measured, like litter and soil C, given assumptions regarding their mean residence times. For instance, in their global synthesis of forest C stock changes, Pan et al. (2011) used simple fractions of growing stocks to estimate soil carbon changes. In national inventories, more detailed models of soil C change can be used.

C stock changes are assumed to be the sum of NEE and lateral C fluxes exported from or imported into the territory considered. For RECCAP2, this territory is the area of each region, where the lateral fluxes consist of C exported to the ocean via inland waters, and exported or imported from trade routes, as it is impractical to have observation-based gridded datasets of lateral fluxes at sub-regional resolution. Therefore, when comparing observation-based C-stock change estimates with independent NEE estimates, e.g. from inversions or other sources, it is strongly recommended to first correct the stock change from each region by the net import or export of C in trade and by the export in rivers. In RECCAP2, there is potential to use smaller sub-regions than in RECCAP1, so that some regions may also receive incoming C, in rivers entering their territory.

### 3.2 Eddy covariance networks

Eddy-covariance flux tower networks measure the net $CO_2$ flux of terrestrial ecosystems (NPP-SHR) across a global network with a typical footprint of about 1 km$^2$. The networks currently consist of about 600 sites (Jung et al., 2020). Given the small footprint, flux tower sites do not adequately measure the fluxes of $F_{geological}$, $F_{fires}$, $F_{reduced}$, $F_{rivers + lakes\ outgas}$ (except for a very few towers in wetlands or flooded systems), $F_{crop\ products}$ and $F_{wood\ products}$. For $F_{grazing}$, only the fraction emitted as $CO_2$ by livestock in the field (not in the barn) in the footprint of a tower is measured. Too few towers are installed over ecosystems in





transition at different times after a land use change, and the network is potentially biased toward younger, more productive,
forest stands, so that regional estimates of $F_{LUC}$ cannot be directly obtained from eddy-covariance flux towers measurements.
The small spatial footprint of eddy flux towers can be up-scaled into gridded maps of NPP-SHR (NEE at ecosystem level)
using the relationship between the continuous measurements from flux towers and simultaneously recorded climate and
vegetation parameters. The fluxes are up-scaled using gridded predictors from remote sensing (such as FAPAR or NDVI) and
climate fields, using machine learning or data-assimilation techniques (Jung et al., 2020; Tramontana et al., 2016).


Both inventories and eddy covariance networks provide point sampling with many gaps between points. These gaps are filled
using up-scaling models like FLUXCOM (Jung et al., 2020; Tramontana et al., 2016). The FLUXCOM data show fair
agreement with inversions and TRENDY models for the seasonal cycle of NEE and for the phase of inter-annual NEE
anomalies (Jung et al., 2017) but the absolute magnitude of interannual anomalies is strongly under-estimated. One attempt to
close the global NEE budget by combining FLUXCOM estimates of NPP – SHR with other fluxes not measured by flux towers
(Zscheischler et al., 2015) obtained a net sink of $CO_2$ larger by 10 PgC y$^{-1}$ than the net land $CO_2$ sink deduced from the global
budget. One possible reason for this mismatch could be biases introduced during the processing of micro-meteorological
observations, for instance u* filtering, or the sampling bias in the tower network. The tower sites are not randomly distributed,
and therefore measure fewer recently disturbed ecosystems (typically C sources) than recovering ones (C sinks), thus
overestimating $CO_2$ uptake given the available network. Since we do not know the true distribution of land fluxes, up-scaling
models of flux towers data could miss important ecosystems not sampled by the training data, or representative landscape
elements with intense sources (peatlands, permafrost, disturbed ecosystems) or sinks (peatlands, plantations) that might
contribute significantly to the carbon balance of a region.

We recommend that RECCAP-2 teams use eddy covariance estimates of net ecosystem $CO_2$ fluxes, but since they consist only
of NPP – SHR, these fluxes should add other $CO_2$ fluxes that are not measured by this approach. This can be done using
aggregated estimates of the non-measured C fluxes in each region, or using gridded estimates. For instance, Zscheischler et al.
(2015) used gridded estimates of $F_{fires}$, $F_{rivers + lakes\ outgas}$, $F_{LUC}$, $F_{crop\ products}$ and $F_{wood\ products}$. They did not add $F_{reduced}$ but gridded
monthly estimates of this flux could be included in RECCAP2 based e.g. on Guenther et al. (2012). We should remain cautious,
noting that NPP–SHR upscaled from eddy flux towers so far gives unrealistically high global $CO_2$ sinks.

**3.3 Other ground-based measurements**

The list provided here is not exhaustive. It includes 'ecological' measurements of NPP e.g. (Olson et al., 2001), biometric C
stock changes at site level, e.g. (Campioli et al., 2015; Luyssaert et al., 2007), soil respiration e.g. the SRDB database (Bond-
Lamberty and Thomson, 2010) and pCO₂ data in rivers and lakes (GLORICH). These measurements are sparse and local in
nature. In a similar fashion to the flux towers measurements described above, it is possible to derive empirical relationships
linking point data with local climate and other predictor variables; these relationships can then be used for spatial or temporal



extrapolation using gridded fields of the same predictors. In recent years, gridded estimates have been provided for soil respiration (Hashimoto et al., 2015) and soil heterotrophic respiration (Konings et al., 2019; Tang et al., 2019), for $F_{river+lakes\ outgas}$ (Lauerwald et al., 2015; Raymond et al., 2014), which can be used to create regional totals.

**3.4 Models driven by statistical data**

Here we refer to a variety of models that do not use physical measurements at selected locations, but rather statistical data about harvested C, C in product pools, and C traded or consumed. These data are usually sourced from national or international statistical agencies or sector bodies. Examples are the study of Wolf et al., (2015) who estimated crop NPP, $F_{grazing}$ and $F_{crop\ products}$, Krausmann et al. (2013) who estimated crop NPP from statistical data on yield, Ciais et al. (2007) who estimated $F_{crop\ products}$

and the corresponding $CO_2$ uptake by growing crops and horizontal displacement of harvested crop biomass, and Zscheischler et al. (2015) who provided gridded estimates of $F_{wood\ products}$ albeit ignoring trade.

**3.5 Models driven by satellite data**

Satellite data are also used in up-scaling forest inventory, eddy covariance and other ground-based measurements, although giving a full list of this category of models is not the purpose of this paper. Here we refer to satellite-driven NPP models

(Bloom et al., 2016; Kolby Smith et al., 2015; Running et al., 2004; Tum et al., 2016; Wißkirchen et al., 2013b) based on light use efficiency formulations, or hybrid land carbon-cycle models that explicitly represent photosynthesis (and NPP), driven by directly-assimilated satellite data. Similarly, fire emission models like GFED and GFAS rely on satellite input data like burned area and fire radiative power (FRP) but estimate emissions using fields from models or other datasets (information on the fuel load, the burning completeness, and emission factors for different gaseous species). Remotely-sensed models of above ground

biomass, derived from optical sensors, i.e., MODIS (Baccini et al., 2017), lidar from ICESAT-1 GLAS (Saatchi et al. 2011), synthetic aperture radar or SAR (Santoro et al. 2018), and L-band vegetation optical depth (VOD, Liu et al. 2015) have been produced globally and regionally (i.e., for mangroves using X-band radar, Simard et al., 2019). when they are repeated over time allow estimates of biomass stock change, such as those presented by Brandt et al. (2018) over Africa and Fan et al. (2019) over the tropics. These datasets differ not only by methodology and training datasets, but by spatial resolution (300 meter to

25 km), and by temporal resolution (annual, or epoch), and so an ensemble-based approach is preferable for assessing uncertainty. Below-ground carbon stock estimates are more challenging to access, for live root biomass often a scaling assumption is made, but for mineral and organic carbon, estimates are derived from empirical upscaling or inventory approaches or process-based models described in Section 3.6.

**3.6 Process-based terrestrial carbon cycle models**

Dynamic global vegetation models (DGVM) simulate bottom-up NEE and a number of ecosystem carbon pools and fluxes, and their change over time on a gridded basis worldwide. The grid resolution ranges from 0.5° for global applications e.g. TRENDY (Sitch et al., 2015) or MstMIP (Wei et al. 2014) to fine resolutions (300 m or less) regionally. These models are not





tightly driven by observations (unlike those in 3.5); but some observations are used by modelers to calibrate parameters. TRENDY models now are benchmarked following ILAMB (Friedlingstein et al., 2019). Dense observation datasets are not

assimilated systematically, although some Carbon Cycle Data Assimilation Systems exist that make use of DGVMs (Kaminski et al., 2013; MacBean et al., 2016) or simpler models like CARDAMOM (Bloom et al., 2016). The advantages of DGVM models for carbon budgeting are that: 1) they provide ensemble of gridded NEE and NEE component estimates as part of TRENDY, 2) these models should in principle conserve mass and simulate consistent C fluxes and C stock changes for all regions. A limitation of DGVMs apart from the fact that they can differ substantially from observations, is that they do not

explicitly represent some of the fluxes in Fig 2. $F_{fires}$ is available from 10 out of 16 DGVM models in TRENDY and FIREMIP (Hantson et al., 2020). $F_{LUC}$ from DGVMs includes a foregone sink of $CO_2$ called the Loss of Additional Sink Capacity (Gasser et al., 2020; Pongratz et al., 2014) which is not included in data-driven methods to quantify this flux (see Section 4). DGVM models partly include $F_{wood\ products}$ and $F_{crop\ products}$ but assume that all harvest is released locally as $CO_2$ to the atmosphere, ignoring lateral displacement of harvested C within and across regions. DGVM models ignore $F_{reduced}$ and only one or two

include $F_{rivers\ +lakes\ outgas}$. Hence, care should be taken when combining DGVM models output with observation-based estimates of C fluxes because of double counting or undercounting. For instance, C outgassing from rivers and lakes derives from C exported by soils, but if this export is not represented in a DGVM, C will be otherwise released as SHR, so that adding to DGVM output an outgassing flux would lead to an erroneous double accounting.

In general, for RECCAP2, we recommend to describe exactly what each estimation approach includes or excludes, for each C flux of Fig. 2, in order to minimize the risk of missing some fluxes or double counting others. Mass conservation should be the key underlying principle when combining bottom up C fluxes originating from different approaches.

## 4 Fluxes from land-use change

Fluxes from land-use change and management (abbreviated to $F_{LUC}$ and defined as having a positive sign for net fluxes from

the atmosphere to the land C) are defined as changes in C-stocks due to deforestation, forest degradation and afforestation or reforestation, wood harvest, subsequent regrowth of forest following harvest or agriculture abandonment, conversion between croplands and grasslands (also sometimes called pastures, or more generally, rangelands), as well as management practices such as shifting cultivation (land cyclically rotating between forest and agriculture). Where applicable, peat burning and drainage should also be considered, as well as carbon fluxes related to management practices such as fire management,

particularly if those practices have changed within the relevant period (for instance, when historically burning ecosystems are subject to fire suppression, or where fire-protected ecosystems become fire-susceptible) (Alvarado et al., 2020; Forkel et al., 2017; Kelley et al., 2019). Where possible, $F_{LUC}$ should be separated into the component fluxes corresponding to the different processes and adding up to the net regional $F_{LUC}$. Typical components of $F_{LUC}$, as reported by bookkeeping models, include immediate biomass losses during deforestation, delayed emissions from soil carbon and litter decomposition for all subsequent





years, following land use change (legacy emissions), emissions from wood products harvested as a result of deforestation, or

derived from secondary forests, and recovery gains due to secondary forest regrowth or afforestation (Hansis et al., 2015;

Houghton et al., 2012). Previous versions of the Houghton et al., (2012) bookkeeping model (up until 2017) reported emissions

from shifting cultivation as part of $F_{LUC}$, but this term has been dropped in the most recent version of this model (Houghton

and Nassikas, 2017). Houghton and Nassikas (2017) also provide emissions from forest degradation (i.e. biomass-reducing

activities that do not result in the land parcel being reclassified as a non-forest) and subsequent recovery as part of $F_{LUC}$.

The various methods available to quantify $F_{LUC}$ (Table 1) rely on different input datasets and models with different abilities to

represent land use practices. They further use different terminology and assumptions on which component fluxes to include,

leading to inconsistencies between one another. For RECCAP2, the best data available in each region should be used. However,

it is crucial to define clearly the methods and assumptions made, and which $F_{LUC}$ fluxes are included in the corresponding

results. If possible, regional $F_{LUC}$ fluxes estimated by the "best method" should be compared with those estimated by the global

datasets from the most up-to-date GCP-Global Carbon Budget in order to ensure consistency and comparability between

regions. The methods used to estimate $F_{LUC}$ include: (i) Bookkeeping models (BK), (ii) Dynamic Global Vegetation Models

(DGVM), (iii) Remote-sensing based methods, (iv) National inventories as detailed below.

**4.1 Bookkeeping models**

Bookkeeping models rely on present-day vegetation and soil C-densities (aggregated or spatially-explicit) and different

response curves (i.e. time courses of change) to estimate changes in C-stocks following a given transition. The two BK models

used in the Global Carbon Budget (GCB) (Friedlingstein et al., 2019) are those from Houghton and Nassikas (2017) and Hansis

et al. (2015), referred to as H&N and BLUE respectively. Both BK models are able to provide $F_{LUC}$ at country-level, but differ

in a number of characteristics, such as the input data, the C-densities and response curves used, the spatial resolution and period

covered, as summarized in Table 1. Spatially-explicit BK models such as BLUE can be adapted to run at regional scales with

finer spatial resolution of land-use change, derived from either national inventories or from remote-sensing (RS) based

transitions (e.g. ESA-CCI Land-cover). If very good data on C-densities and, ideally, response curves, is available regionally,

and no superior regionally BK model is available, BLUE can also be adapted to run with that information at country or regional

level.

**4.2 Dynamic Global Vegetation Models (DGVMs)**

DGVMs explicitly simulate the processes controlling photosynthesis, growth, decomposition and mortality of vegetation and

the processes involved in soil-C changes. They also simulate the fluxes resulting from forest clearing, pasture and crop

conversion, abandonment and re-growth and crop harvest, although the implementation varies between models, as do the

assumptions about the areas being converted (e.g. gross versus net conversion, see section 4.5), the management practices

included, and the fate of C following transitions. DGVMs in RECCAP2 can be used to estimate $F_{LUC}$ in two ways: (i) the





global simulations from TRENDY for GCB2019 can be analyzed at country or region level; and (ii) any DGVM including the aforementioned processes can be forced with better or finer data at country or regional level. If a DGVM with an improved representation of regional processes is available, it is recommended to use it rather than more generic global models. Yet, it is
important for regional models to follow the simulation protocols of TRENDY (Friedlingstein et al., 2019) to facilitate comparison between regions. In order to estimate $F_{LUC}$ with DGVMs, factorial simulations with and without LUC from the pre-industrial period until present are generally used. The year 1700 should be used as the reference data for the pre-industrial state in RECCAP2, in order to be consistent with the TRENDY protocol in depicting legacy fluxes.

There are different ways to estimate $F_{LUC}$, which partly explain differences between DGVMs and BK models. The DGVM simulations used to evaluate $F_{LUC}$ under different assumptions are listed in Table 2. Up to now, $F_{LUC}$ from DGVMs has been estimated from the difference between two simulations, one forced with changing $CO_2$, climate and LUC, and another forced with changing $CO_2$ and climate but a fixed pre-industrial land-cover map (corresponding to S2-S3 in the TRENDY protocol). The potential natural vegetation in the simulation with fixed land cover (S2) is affected by $CO_2$ fertilization and therefore
provides an additional sink that is lost e.g. when deforestation occurs. This foregone sink is Loss of Additional Sink Capacity (LASC) (Gasser et al., 2020; Pongratz et al., 2014). For consistency with BK models, $F_{LUC}$ estimates with no LASC and for present-day C-densities should be delivered instead, based on differences between two simulations under time-invariant present-day environmental conditions of climate, $CO_2$, N-deposition and N-fertilization: one with LUC (S5 in the TRENDY protocol and Fig. 3) and one with fixed pre-industrial (1700) land cover (S6 in the TRENDY protocol). In that case, $F_{LUC}$ can
be estimated as:

$$F_{LUC\text{-}S5} = S5 - S6 \tag{1}$$

Because $F_{LUC}$ from both S5 and BK models are forced with present-day C-densities which have on average increased during the perturbation of the carbon cycle since pre-industrial times, they may overestimate LUC emission fluxes in the first part of the last Century. Therefore, an additional simulation (S4) can be performed, where models are forced with time-invariant pre-
industrial environmental conditions, and annual time-varying land use 1700-2018. In that case,

$$F_{LUC\text{-}S4} = S4 - S0 \tag{2}$$

where S0 is a control simulation with time-invariant pre-industrial (1700) $CO_2$, climate, and land use. In this case, $F_{LUC}$ is calculated based on pre-industrial potential C densities and does not include LASC. For consistency, the natural land sink over areas not affected by LUC can then be estimated with DGVMs as:

$$S_{LAND} = S3 - S0 + S6 - S5 = S3 - S4 \tag{3}$$

For RECCAP2, we recommend that $F_{LUC}$ from DGVMs is estimated following Eq. 1 ($F_{LUC\text{-}S5}$) so that results can best be compared with BK results in the recent decades.





## 4.3 Remote-sensing data related to LUC

Several global remote-sensing products can be useful in estimating $F_{LUC}$ in RECCAP2. They can be applied in various ways:
1) Estimate land-cover change in the recent decades to produce regional transition maps at finer spatial-scale and with better accuracy than are currently available. These maps can then be used to force BK or DGVMs, 2) provide finer-resolution and globally consistent maps of vegetation C-densities (for undisturbed locations) that can be used in BK models, 3) directly estimate changes in biomass C-stocks, for instance using optical data (Harris et al., 2012) vegetation-optical depth (Fan et al., 2019) or Lidar data and report these only for deforestation areas (to exclude environmentally-induced fluxes).

Examples of already available remote sensing-based datasets than can be used for land-cover and land cover change mapping are the ESA-CCI Land-Cover product, based on five different satellite missions, at 300m spatial resolution and annual time-steps between 1992 and 2018 (Santoro et al., 2017), the Landsat 30m spatial-resolution forest cover change product covering 2000 to 2018 (Hansen et al., 2013) extended to land cover change for forest, short vegetation and bare soil (Song et al., 2018).
For vegetation C-densities, the ESA GlobBiomass dataset provides above-ground biomass data for a period centered on the year 2010 at 100 m spatial resolution (Santoro, 2018). Because of its fine spatial resolution, this dataset could in principle, be used to evaluate undisturbed C-densities (Erb et al., 2018; Luyssaert et al., 2012). Other datasets currently under development include the ESA-CCI high-resolution Land-cover, expected to provide a long-term record since 1990s of regional high-resolution land cover maps at 30 m spatial resolution every 5 years in regions of interest, the ESA-CCI Biomass dataset, which
will provide above ground biomass data for four epochs of mid 1990s, 2007-2010, 2017/2018 and 2018/2019 at 100 m spatial resolution with a relative error of less than 20%. The NASA Carbon Monitoring System program is also supporting the development of regional to global scale biomass products based on optical reflectance data from MODIS, as well as active lidar-based approaches using ICESAT-1 and now ICESAT-2 GLAS-retrievals, and the Global Ecosystem Dynamics Instrument or GEDI aboard the International Space Station. The lidar approaches require integration with wall-to-wall optical
measurements as lidar is a 'shot' retrieval with a fairly small footprint size, but with high-accuracy in terms of measurement ability to retrieve canopy height and thus biomass (Dubayah et al., 2020). Satellite-based products have important advantages for estimating contemporary direct emissions from changes in aboveground biomass, such as global coverage, consistency, reliability, and increasingly-higher spatial resolution. However, they cannot estimate legacy soil fluxes from land-use change prior to the satellite era, nor are they able to separate the contribution of environmental changes on FLUC. The comparison of
FLUC derived from RS-based methods with DGVMs or BK estimates should therefore be made with care.

## 4.4 National inventories

National Greenhouse Gas Inventories (NGHGI) report anthropogenic emissions and sinks to the UNFCCC, and are the official numbers used to take stock of the Nationally Determined Contributions (NDCs). NGHGI use different definitions and assumptions than those used by the carbon-cycle research community, as detailed in (Grassi et al., 2018). NGHGI, in their



Agriculture Forestry and Land Use (AFOLU) sector, report $CO_2$ fluxes of land under management, as defined by each country. Such managed land can include areas under nature conservation management. The C balance of established cropland, grassland and forests are reported from national inventories, under the LULUCF sub-sectors, including C fluxes of transitions involving managed lands. A variety of approaches are used by NGHGIs, mostly based on general emission factors following IPCC guidelines. Only lands converted within the past 20 years are included under LULUCF fluxes, unlike BK and DGVM models

that land use change fluxes since 1700 or 1850. On the other hand, NGHGI include land use change fluxes for transitions that are usually not implemented in BK and DGVM models, such as from peatland converted to agriculture and from land converted to human settlements.

## 4.5 Land-use change transitions, definitions and assumptions

The land-use change transitions and land-management fields used in the latest version of the GCP-Global Carbon Budget

(Friedlingstein et al., 2019) to calculate the net land use change flux called $F_{LUC\,latest}$, are from the harmonized land-use change data (LUH2v2.1h) dataset (Hurtt et al., 2011), which is based on HYDE3.1 (Klein Goldewijk et al., 2011). These data have the advantage of being globally consistent and covering a long period (850 - present), but have relatively coarse spatial-resolution ($0.25 \times 0.25$-degree) and, due to a globally consistent methodology, may not account for regional specificities (Bastos et al., 2018; Li et al., 2018). For each region, the best available information (in terms of spatiotemporal resolution or

detail of processes covered) on land-use change should be used. This can be from national statistics, inventories or remote-sensing. In RECCAP2, each regional team will decide the land cover classification scheme that best fits a given region, but it is recommended that the LUH2v2h forest/non-forest distinction be used when classifying rangelands.

## 5 Concluding remarks

We present a way forward for developing consistent top-down and bottom-up estimates for regional carbon dioxide budgets.

The methodology focuses on reconciling the treatment of non-$CO_2$ emissions from $CH_4$, CO, and BVOCs and their contribution to $CO_2$ via atmospheric chemistry, and the treatment of lateral fluxes of carbon. Given the complexity of this task, the approaches toward implementation can be considered using the Tiered approach of the IPCC, whereby higher Tiers use progressively more complex, regionally and locally calibrated sources of information. For example, a Tier 1 approach combines global emission-factors with activity data to estimate fluxes, Tier 2 might use regionally-calibrated emission factors,

whereas Tier 3 uses locally calibrated emission factors to estimate fluxes from activity information. The Global Carbon Project now conducts greenhouse gas budget accounting for the three major greenhouse gases, carbon dioxide, methane, and nitrous oxide, where each budget provides detailed sectoral information for sources and sinks using what is more closely aligned with Tier 1 approaches. Beginning with Tier 1 data can help initiate regional budgets, and identify areas of uncertainty or opportunities for regionally and locally calibrated approaches to be used to reduce uncertainty.



**Code and data availability**

There is no code associated to this paper. RECCAP-2 data will be available to the community after submission of each regional chapter to a peer reviewed journal.

**Author contributions**

P.Ciais designed and wrote the study. F. Chevallier provided input to the inversion section and A. Bastos and J. Pongratz
contributed the land use section. Other contributors helped to improve the text in their field of expertise or globally. H. Yang additionally helped with the references.

**Acknowledgements**

P.Ciais acknowledges funding from the ANR CLAND Convergence Institute. A.Bastos, F. Chevallier and P. Ciais acknowledge support from the VERIFY H2020 project and the RECCAP2 ESA CCI project



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

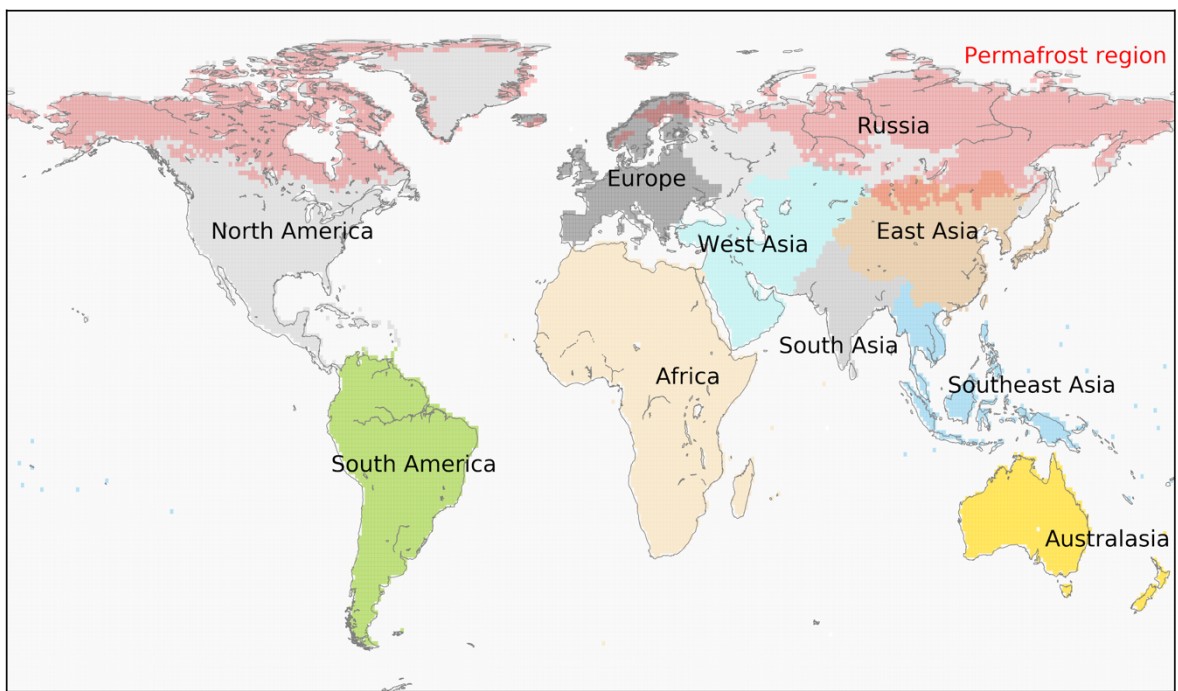

**Figure 1: Map of the RECCAP2 regions.** The 'region' in shaded red corresponds to permafrost covered areas. Map plot courtesy of Naveen Chandra (NIES/JAMSTEC). Files can be accessed at https://ebcrpa.jamstec.go.jp/~prabir/data/region_masks/RECCAP2_Mask11r.nc
https://ebcrpa.jamstec.go.jp/~prabir/data/region_masks/RECCAP2_peramfrost.nc







**Figure 2: Summary of C fluxes to be reported in each RECCAP2 region (top) and name of each flux (bottom).**



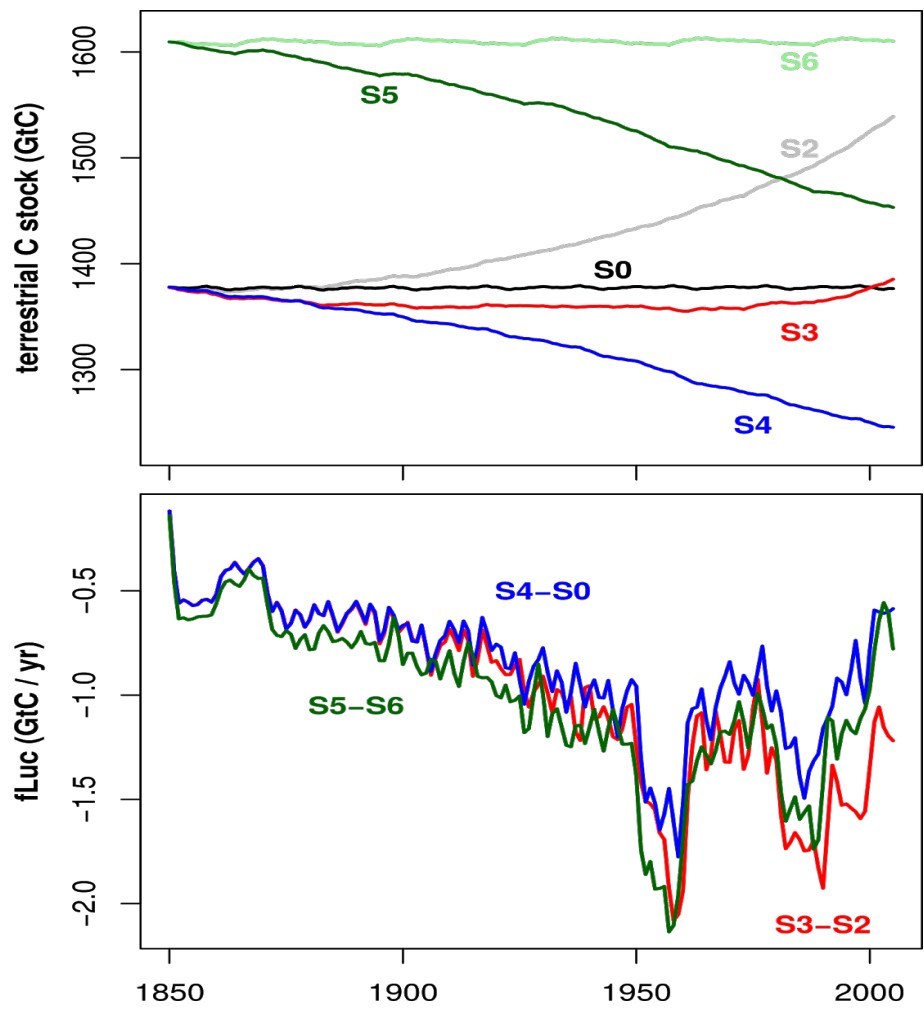

**Figure 3: Terrestrial cumulative C stocks (top) and corresponding $F_{LUC}$ (bottom) as simulated by the JSBACH dynamic vegetation model for the different simulations discussed.** Shown is $F_{LUC}$ derived as S5 minus S6, S3 minus S2, and S4 minus S0 (see text).





**Table 1: Main differences between the methods used to estimate $F_{LUC}$.**

| Method | C-densities | Transitions | Response of C-stocks | Spatial scale | Period covered | Land use practices included | LASC | Legacy fluxes |
|---|---|---|---|---|---|---|---|---|
| H&N Houghton and Nassikas 2017 | Present day, aggregated per biome at country-level from FAO | FAOSTAT (2015) data of croplands and pastures areas since 1961. FAO (2015) changes in forest areas since 1990. Data are provided at country-level, and then decomposed per biome based on MODIS IGBP classification (Friedl et al., 2002). | Response curves from literature | Country to global | 1850 - 2015 | Deforestation; cropland/grassland abandonment; forest degradation; wood harvest; fire suppression (U.S. only); peat drainage and peat burning | No | Yes (for post-1850 LUC) |
| BLUE | Present day, spatially explicit fields per biome type | Spatially-explicit transitions from LUH2v2 <br><br> Can be used with RS-based transition maps | Response curves from literature | 0.25x0.25 lat. Lon. global coverage <br><br> Can be adapted to any other spatial resolution if required | 850 - present (2018 for Friedlingstein et a., 2019) | Deforestation (separated by transitions to cropland and to grassland); cropland/grassland abandonment; forest degradation; wood harvest; shifting cultivation; peat drainage and peat burning | No | Yes (for post-850 LUC) |



| DGVMs S3 - S2 | Transient (climate + $CO_2$) | Spatially-explicit transitions form LUH2v2 | Simulated implicitly by DGVMs based on environmental conditions | Model grid scale, global coverage | 1700 - present (2018 for Friedlingstein et al., 2019) | Model-dependent (see Friedlingstein et al., 2019 for TRENDY models) | Yes | Yes (for post-1700 LUC) |
|---|---|---|---|---|---|---|---|---|
| DGVMs S4 - S0 | Pre-industrial (1700) spatially explicit fields Simulated by DGVMs | Spatially-explicit transitions form LUH2v2 | Simulated implicitly by DGVMs based on environmental conditions | Model grid scale, global coverage | 1700 - present 2018 for Friedlingstein et al., 2019) | Model-dependent (see Friedlingstein et al., 2019 for TRENDY models) | No | Yes (for post-1700 LUC) |
| DGVMs S5 - S6 | Present-day (1999-2018) spatially explicit fields simulated by DGVMs | Spatially-explicit transitions form LUH2v2 | Simulated implicitly by DGVMs based on environmental conditions | Model grid scale, global coverage | 1700 - present 2018 for Friedlingstein et al., 2019) | Model-dependent (see Friedlingstein et al., 2019 for TRENDY models) | No | Yes (for post-1700 LUC) |
| RS-based | | Changes in LUC derived from RS, e.g. ESA-CCI Land-cover or Hansen et al. (fluxes associated with forest change only) | Estimated by BK or calculated directly from RS-based biomass estimates (e.g. ESA-CCI Biomass product) | Sensor-dependent | Satellite record 1980s - | | Yes | Yes (BK) / No (RS only) |





**Table 2: DGVM simulations to calculate $F_{LUC}$ from TRENDY-v8 protocol (Friedlingstein et al. 2019).**

| Simulation | Environmental conditions | Land-cover |
|---|---|---|
| S0 | Time-invariant pre-industrial | Time-invariant pre-industrial |
| S2 | Historic | Time-invariant pre-industrial |
| S3 | Historic | Historic |
| S4 | Time-invariant pre-industrial | Historic |
| S5 | Time-invariant present day | Historic |
| S6 | Time-invariant present day | Time-invariant pre-industrial |