# Peer review of "Definitions and methods to estimate regional land carbon fluxes for the second phase of the REgional Carbon Cycle Assessment and Processes Project (RECCAP-2)"

_Geoscientific Model Development, 2020_

## Short Comment (SC1) · 27 Oct 2020

Dear authors,

usually we do not accept papers with code or data availability sections feeding hopes of future publication.

Therefore, either wait until the data your research is based on is published or at least provide much more information about where the data will be available. (e.g. which special issue in which journal. where will the data be hosted, etc. etc.)

[Figure]

Best regards,

Astrid Kerkweg (Executive Editor)
* * *

---

## Short Comment (SC2) · 27 Oct 2020

Ironically, it has always been easier to construct a global carbon budget than for any other unit of land, whether a region or a hectare. The reason is because of lateral transport of carbon by animals moving between land units, carbon transported by rivers (and not only the atmosphere), and crop and wood products transported by trade. Another troublesome issue for terrestrial carbon budgets relates to the various forms carbon may take, including BVOCs, methane, carbon monoxide. And, in addition to these real-world fluxes, there are the usual scientific issues related to different methods of

measurement. This paper by Ciais et al. looks at the definitions and methods needed to construct regional carbon budgets. An initial REgional Carbon Cycle Assessment and Processes Project (RECCAP) was carried out by the Global Carbon Project for the period 2000-2009. This paper sets the ground for a second: RECCAP-2. The paper discusses a series of issues and provides recommendations for use of transparent, if not identical, methods. One goal is to have the information necessary for reconciling top-down (inverse analyses) with bottom-up (inventory and modeling) approaches for measuring terrestrial carbon fluxes. No question, both the field of terrestrial carbon and the methods available for measuring and inferring fluxes are becoming more and more sophisticated and detailed. This paper seeks to define processes and reconcile different methods of measurement. It is a valuable contribution, not just to terrestrial carbon science, but to preparing for RECCAP-2. There may be nothing new here, but there is a careful review and consolidation of what's needed going forward for transparency and consistency. The paper is comprehensive, well organized and clearly written. I have no criticisms of the work, no suggestions for revision. I would note, however, that although one of the goals of terrestrial carbon research has always been to separate fluxes driven by anthropogenic, as opposed to non-anthropogenic (environmental) processes, that goal has arguably been "dumbed-down" (subverted?) by the IPCC's introduction of the "Managed Land" proxy. National greenhouse gas inventories are included briefly near the end of this paper, but they are likely to require considerable future work to be reconciled with the results from regional carbon budgets as proposed here. That's work for future analyses.

---

## Referee Comment (RC2) · Anonymous Referee #2 · 6 Jan 2021

This manuscript presents a conceptual and methodological framework for the computation of land-atmosphere fluxes in the context of the RECCAP-2 project. It describes in detail the main fluxes to be consider for obtaining the net exchange of carbon between land and atmosphere, with special emphasis in the homogenization of top-down versus bottom-up estimates. The manuscript is well written, and it is a meaningful contribution to the literature. Given that the definitions and conceptual framework described here has applications mostly for the RECAPP project, the manuscript could be publish in its current form after minor revisions. However, if this work is intended to transcend

RECAPP, and provide a useful conceptual framework for global carbon cycle science, then a major revision is required. I have three main concerns that I will detail below, followed by a list of small minor issues.

**1 Major concerns**

- From my point of view, the definition of the main component fluxes of the budget presented in this manuscript, and summarized in Figure 2, mixes two different aspects of a carbon budget. On the one hand, many of the fluxes are defined by the specific process that generate a transfer of carbon from one pool to another (lateral transfers), or from a pool to the atmosphere. This definition of fluxes is intuitive and is a good approximation to our scientific understanding of the main processes in the Earth system that produce transfers of carbon among reservoirs. On the other hand, some of the fluxes, and in particular those related to the land use component, are defined based on the proximate cause of anthropogenic emissions. I think this mix on the way the fluxes are defined is confusing and prone to double counting or confusing accounting. For example, a process that generates emissions of carbon to the atmosphere from the land is the respiration of heterotrophic organisms, which includes wild and domesticated animals as well as humans. Heterotrophic respiration is the main biological process that produces the emission, but one could attribute these emissions based on the type of heterotrophic organisms that produce them. In other words, one can define the flux based on the process (heterotrophic respiration) or based on the proximate cause, e.g. 'carbon emissions from crop biomass consumed by animals and humans' as defined in section 2.5.1. However, it is confusing to define fluxes based on processes and based on proximate causes as part of the same budget. It can also lead to double counting. The same problem appears in the definition of fluxes due to fires and those due to land use change or deforesta-

tion. The process that defines the flux is fire, but the proximate cause may be due to crop management, deforestation, or annual natural disturbances. The fluxes considered in this manuscript is a mix of both, fluxes defined by processes and fluxes defined by the proximate cause. I do not think this can help us to get some clarity in constraining the global carbon budget and to understand its change. A better approach would be to define all fluxes based on the processes that lead to the flux, or to define them based on all the different proximate causes. The idea is to be consistent. I personally would be prefer definitions of fluxes based on processes, and in a posteriori analysis, attribute the fluxes to specific proximate causes. I think such an approach would help to get separate two main aims in current C cycle research, to understand processes, and to attribute causes of change.

- Although the aim of this project is on the fluxes of carbon between the atmosphere and land, it is surprising that no effort is placed in quantifying and reporting carbon stocks of the main source pools from the land. Knowledge on the carbon stocks is important for two main reasons: 1) to know the relative proportion of carbon emitted from source pools and how they differ among main regions, and 2) to identify potential mass balance problems when fluxes are much larger than the size of the source pool. For reporting based on Delta methods, reporting the size of the pools is easy and should be recommended.

- The recommendation of reporting NPP instead of GPP is troubling, and does not reflect well our current physiological understanding of carbon assimilation in terrestrial ecosystems. The authors define NPP as 'the flux of carbon transformed into biomass tissues after fixation by GPP', probably assuming that autotrophic respiration is already removed in NPP; i.e. NPP = GPP - autotrophic respiration. The problem with this definition is that we know that autotrophic respiration can only occur from living tissues produced after biomass formation, not before as the definition implies. Only living cells can respire carbon, and experiments and

isotopic analyses have shown that carbon respired from roots and stems can be years to decades old. While GPP quantifies the instantaneous removal of CO2 from the atmosphere, autotrophic respiration is the lagged release of CO2 back to the atmosphere. These fluxes are not necessarily in sync, and therefore NPP is a poor approximation of the instantaneous net flux. This is important for the planned comparison of fluxes from the inversions, because they are computed at much higher temporal resolutions than the NPP estimates from forest inventories. In addition, there are now a range of techniques that aim at quantifying GPP in ecosystems using measurements of fluorescence and COS both from satellites and at flux tower sites. Therefore, there is an opportunity to include independent estimates of carbon assimilation (GPP) as part of the regional carbon balances.

**2   Minor comments**

- Line 62. Add 'of' after 'estimates'

- Ln 190. Add ',' or ';' after 'regions'

- Ln 224. Add 'be' in 'needs to paid'

- Section 2.2.3. The quantification of carbon fluxes due to trade is interesting. Would it be useful to include also carbon fluxes due to trade of unburned fossil fuels?

- Line 371. This sentence is similar to line 360 in previous paragraph. Consider removing it.

- Line 614. NEE at the ecosystem level only considers CO2, at least as it is commonly done in eddy-covariance studies. However, you define regional NEE as the net carbon balance of carbon, not just CO2. I see a mismatch here between

the more traditional definition of NEE at the ecosystem level and your new definition at the regional level. Wouldn't be better to call your new quantity net regional carbon balance?

- Figure 2. I don't see the benefit of repeating the same figure twice to only add the names of the fluxes. I would make only one diagram with the abbreviations and define the flux names in a table.

---

## Author Comment (AC4) · 13 Jun 2021

Reply to editor

Many datasets we will work with are publicly available (e.g. CMIP6, ocean models, ICOS+FLUXNET, FLUXCOM, LUH2, ...). Most of the global datasets which aren't public yet are already available to the RECCAP2 teams through the MPI-data portal:

https://www.bgc-jena.mpg.de/geodb/projects/Data.php C1
These can be made publicly available at a later stage, but in any case, RECCAP2 studies will be hosted as a Special Collection at AGU. Following AGU data policy, the datasets used in each paper will be made available upon publication.

https://www.agu.org/Publish-with-AGU/Publish/Author-Resources/Policies/Data-policy

We will encourage teams to do this through the ICOS carbon portal (https://www.icos-cp.eu). In any case, the fluxes that we discuss in the paper go way beyond RECCAP2 and are used commonly in the C cycle community from different datasets, not only those we plan to use. So the paper's guidelines are relevant, even if the particular data used in RECCAP2 is not associated with this manuscript

---

## Author Response (AR1)

**Response to referees comments on "Definitions and methods to estimate regional land carbon fluxes for the second phase of the REgional Carbon Cycle Assessment and Processes Project (RECCAP-2)"** *by* **Philippe Ciais et al.**

**Richard Houghton (Referee)**

Ironically, it has always been easier to construct a global carbon budget than for any other unit of land, whether a region or a v hectare. The reason is because of lateral transport of carbon by animals moving between land units, carbon transported by rivers (and not only the atmosphere), and crop and wood products transported by trade. Another troublesome issue for terrestrial carbon budgets relates to the various forms carbon may take, including BVOCs, methane, carbon monoxide. And, in addition to these real-world fluxes, there are the usual scientific issues related to different methods of measurement. This paper by Ciais et al. looks at the definitions and methods needed to construct regional carbon budgets. An initial REgional Carbon Cycle Assessment and Processes Project (RECCAP) was carried out by the Global Carbon Project for the period 2000-2009. This paper sets the ground for a second: RECCAP-2. The paper discusses a series of issues and provides recommendations for use of transparent, if not identical, methods. One goal is to have the information necessary for reconciling top-down (inverse analyses) with bottom-up (inventory and modeling) approaches for measuring terrestrial carbon fluxes. No question, both the field of terrestrial carbon and the methods available for measuring and inferring fluxes are becoming more and more sophisticated and detailed. This paper seeks to define processes and reconcile different methods of measurement. It is a valuable contribution, not just to terrestrial carbon science, but to preparing for RECCAP-2. There may be nothing new here, but there is a careful review and consolidation of what's needed going forward for transparency and consistency. The paper is comprehensive, well organized and clearly written. I have no criticisms of the work, no suggestions for revision. I would note, however, that although one of the goals of terrestrial carbon research has always been to separate fluxes driven by anthropogenic, as opposed to non-anthropogenic (environmental) processes, that goal has arguably been "dumbed-down" (subverted?) by the IPCC's introduction of the "Managed Land" proxy. National greenhouse gas inventories are included briefly near the end of this paper, but they are likely to require considerable future work to be reconciled with the results from regional carbon budgets as proposed here. That's work for future analyses.

**Response:**

We thank the reviewer R Houghton for the summary and appraisal of the manuscript. We agree that the use of 'managed land' as a proxy of 'direct human effect' can be a source of inconstancy between "scientific approaches" and UNFCCC accounting, as noted by Grassi et al. 2017. We added a short section on this issue and recommended to sample if possible gridded estimates (DGVMs, inversions, satellite products) of carbon fluxes over managed land areas, following either country information of (if spatially explicit areas of managed vs non-managed is not provided) to use e.g. the mask of Potapov et al. (2017) for un-managed forests. Our manuscript is focused on CO2 budgets of RECCAP-2 regions and additional work is ongoing to reconcile DGVM, Bookkeeping models, Inversions with National inventories. Following these guidelines, and including the separation of managed vs. unmanaged land, Deng et al. (2021), made a first step in the direction of reconciling top-down

and bottom-up estimates of net CO2, CH4 and N2O fluxes. Their results show that the approach proposed here increases agreement between these estimates and national inventories for several countries.

Potapov, P., Hansen, M. C., Laestadius, L., Turubanova, S., Yaroshenko, A., Thies, C., Smith, W., Zhuravleva, I., Komarova, A., Minnemeyer, S., and Esipova, E.: The last frontiers of wilderness: Tracking loss of intact forest landscapes from 2000 to 2013, Science Advances, 3, e1600821, https://doi.org/10.1126/sciadv.1600821, 2017.

Deng, Z., Ciais, P., Tzompa-Sosa, Z. A., Saunois, M., Qiu, C., Tan, C., Sun, T., Ke, P., Cui, Y., Tanaka, K., Lin, X., Thompson, R. L., Tian, H., Yao, Y., Huang, Y., Lauerwald, R., Jain, A. K., Xu, X., Bastos, A., Sitch, S., Palmer, P. I., Lauvaux, T., d'Aspremont, A., Giron, C., Benoit, A., Poulter, B., Chang, J., Petrescu, A. M. R., Davis, S. J., Liu, Z., Grassi, G., Albergel, C., and Chevallier, F.: Comparing national greenhouse gas budgets reported in UNFCCC inventories against atmospheric inversions, Earth Syst. Sci. Data Discuss. [preprint], https://doi.org/10.5194/essd-2021-235, in review, 2021.

**Response to referees comments on "Definitions and methods to estimate regional land carbon fluxes for the second phase of the REgional Carbon Cycle Assessment and Processes Project (RECCAP-2)"** *by* **Philippe Ciais et al.**

**Anonymous Referee #2**

This manuscript presents a conceptual and methodological framework for the computation of land-atmosphere fluxes in the context of the RECCAP-2 project. It describes in detail the main fluxes to be consider for obtaining the net exchange of carbon between land and atmosphere, with special emphasis in the homogenization of top-down versus bottom-up estimates. The manuscript is well written, and it is a meaningful contribution to the literature. Given that the definitions and conceptual framework described here has applications mostly for the RECAPP project, the manuscript could be publish in its current form after minor revisions. However, if this work is intended to transcend RECAPP, and provide a useful conceptual framework for global carbon cycle science, then a major revision is required. I have three main concerns that I will detail below, followed by a list of small minor issues.

**Response:**

We thank the reviewer for this positive comment and have done our best to address the three issues highlighted below.

**1 Major concerns**

From my point of view, the definition of the main component fluxes of the budget presented in this manuscript, and summarized in Figure 2, mixes two different aspects of a carbon budget. On the one hand, many of the fluxes are defined by the specific process that generate a transfer of carbon from one pool to another (lateral transfers), or from a pool to the atmosphere. This definition of fluxes is intuitive and is a good approximation to our scientific understanding of the main processes in the Earth system that produce transfers of carbon among reservoirs. On the other hand, some of the fluxes, and in particular those related to the land use component, are defined based on the proximate cause of anthropogenic emissions. I think this mix on the way the fluxes are defined is confusing and prone to double counting or confusing accounting. For example, a process that generates emissions of carbon to the atmosphere from the land is the respiration of heterotrophic organisms, which includes wild and domesticated animals as well as humans. Heterotrophic respiration is the main biological process that produces the emission, but one could attribute these emissions based on the type of heterotrophic organisms that produce them. In other words, one can define the flux based on the process (heterotrophic respiration) or based on the proximate cause, e.g. 'carbon emissions from crop biomass consumed by animals and humans' as defined in section 2.5.1.

However, it is confusing to define fluxes based on processes and based on proximate causes as part of the same budget. It can also lead to double counting. The same problem appears in the definition of fluxes due to fires and those due to land use change or deforestation. The

process that defines the flux is fire, but the proximate cause may be due to crop management, deforestation, or annual natural disturbances. The fluxes considered in this manuscript is a mix of both, fluxes defined by processes and fluxes defined by the proximate cause. I do not think this can help us to get some clarity in constraining the global carbon budget and to understand its change. A better approach would be to define all fluxes based on the processes that lead to the flux, or to define them based on all the different proximate causes. The idea is to be consistent. I personally would be prefer definitions of fluxes based on processes, and in a posteriori analysis, attribute the fluxes to specific proximate causes. I think such an approach would help to get separate two main aims in current C cycle research, to understand processes, and to attribute causes of change.

**Response**

We agree that fluxes can be defined by process or by cause. In particular, heterotrophic respiration is a process underlying different fluxes: soil microbial decomposition, oxidation of crop products by animal and human metabolism, decomposition of DOC in river and lakes, and even part of the "land use change" $CO_2$ emissions from slash and legacy soil carbon decomposition. Attribution by cause is mostly beyond the goal of RECCAP-2 because it would require to separate 'direct human induced' and 'indirect effects' rather than considering 'causes' as individual drivers (e.g. fire is a driver but can be of anthropogenic or natural causation, and the state of the science does not allow to properly separate these two causes). This has been a source of inconsistency between carbon models used for IPCC WG1 and WG3 and national reporting for IPCC Guidelines used for UNFCCC reports. See comment by R. Houghton.

We offer to mention the issue of 'direct human induced' and 'indirect effects' but clearly adopt an attribution by 'process' in RECCAP2. Thus, we are left with heterotrophic respiration and combustion which are cross-cutting processes. In our study, the definitions are based on practical methods to estimate each flux. The method used to estimate river outgassing (from river pCO2 data) is, for instance, different from that used to estimate "terra firme" soil heterotrophic respiration (from site data or models), or animal and human digestion by heterotrophic processes (from agricultural and trade statistics). In bookkeeping models for FLUC calculation, respiration fluxes from crop and wood harvest slash and legacy are considered and – when sufficient data is available – they can be separated by cause (e.g. management, deforestation, land-abandonment, shifting cultivation etc). See Hansis et al. (2015) for an example of such decomposition of the different terms. One term that is not considered by bookkeeping models is the effect of climate and environmental variability and change on sinks and sources (including respiration fluxes) resulting from LUC. This term (LASC), is in part incorporated in FLUC estimates by DGVMS, which are also commonly used to estimate FLUC, and may be difficult to separate from natural fluxes. Hence our recommendation to estimate FLUC following Eq. 1.

To address the reviewer's concern, we added a table that gives the possibility to regroup all fluxes belonging to 'heterotrophic respiration' into aquatic C decomposition, intact lands and land in transition soil carbon decomposition, and biomass combustion (biofuels and wildfires).

Although the aim of this project is on the fluxes of carbon between the atmosphere and land, it is surprising that no effort is placed in quantifying and reporting carbon stocks of the main source pools from the land. Knowledge on the carbon stocks is important for two main

reasons: 1) to know the relative proportion of carbon emitted from source pools and how they differ among main regions, and 2) to identify potential mass balance problems when fluxes are much larger than the size of the source pool. For reporting based on Delta methods, reporting the size of the pools is easy and should be recommended.

**Response**

We agree that carbons stock reporting is important, not only stock changes. Some stocks like permafrost carbon, peat, mineral associated soil organic matter, or carbon sediment have a small exchange flux with the atmosphere today, but need to be accounted for future budgets. In the revised manuscript we added as a recommendation for RECCAP2 to report the size of pools and how this size was determined.

The recommendation of reporting NPP instead of GPP is troubling, and does not reflect well our current physiological understanding of carbon assimilation in ter- restrial ecosystems. The authors define NPP as 'the flux of carbon transformed into biomass tissues after fixation by GPP', probably assuming that autotrophic respiration is already removed in NPP; i.e. NPP = GPP - autotrophic respiration. The problem with this definition is that we know that autotrophic respiration can only occur from living tissues produced after biomass formation, not before as the definition implies. Only living cells can respire carbon, and experiments and isotopic analyses have shown that carbon respired from roots and stems can be years to decades old. While GPP quantifies the instantaneous removal of $CO_2$ from the atmosphere, autotrophic respiration is the lagged release of $CO_2$ back to the atmosphere. These fluxes are not necessarily in sync, and therefore NPP is a poor approximation of the instantaneous net flux. This is important for the planned comparison of fluxes from the inversions, because they are computed at much higher temporal resolutions than the NPP estimates from forest inventories. In addition, there are now a range of techniques that aim at quantifying GPP in ecosystems using measurements of fluorescence and COS both from satellites and at flux tower sites. Therefore, there is an opportunity to include independent estimates of carbon assimilation (GPP) as part of the regional carbon balances.

**Response**

We agree that autotrophic respiration occurs necessarily from carbon that has been fixed and (temporarily) incorporated into plants (a small part of that flux can also come from 'older' reserve and labile pools). However, the most relevant variable for annual and decadal budgets is the fraction of plant carbon that enters ecosystems and has a residence time larger than typically one year. Here, NPP, despite definition and measurement issues is a good approximation of this "incoming flux". So we prefer to recommend NPP as a priority and GPP as additional information. This has been changed in the manuscript.

**2 Minor comments**

- Line 62. Add 'of' after 'estimates'
- Ln 190. Add ',' or ';' after 'regions'
- Ln 224. Add 'be' in 'needs to paid'

Changed

- Section 2.2.3. The quantification of carbon fluxes due to trade is interesting. Would it be useful to include also carbon fluxes due to trade of unburned fossil fuels?

Added

- Line 371. This sentence is similar to line 360 in previous paragraph. Consider removing it.

Done

Line 614. NEE at the ecosystem level only considers CO2, at least as it is commonly done in eddy-covariance studies. However, you define regional NEE as the net carbon balance of carbon, not just CO2. I see a mismatch here between the more traditional definition of NEE at the ecosystem level and your new definition at the regional level. Wouldn't be better to call your new quantity net regional carbon balance?

**Response**

We prefer to use NEE-C instead of NEE in the revised manuscript. The difference between NEE "CO2" from eddy covariance data and our definition of NEE-C is clearly explained in the revised manuscript.

• Figure 2. I don't see the benefit of repeating the same figure twice to only add the names of the fluxes. I would make only one diagram with the abbreviations and define the flux names in a table.

**Response**

Thank you for this comment. We modified Fig. 2 as suggested with the flux names in a table

---

## Author Response (AR2)

Reply to the editor

Dear authors,
Thanks for preparing a revised version of the manuscript addressing reviewers' comments. This version addresses well most of those comments, and it can be accepted for publication. The new Table 1 is a very nice addition that addresses well one of the main concerns from Reviewer 2.

Thank you

Before submitting the final revised version, please review once again your text. I think the new text in the Code Availability section and the Author Contributions can be improved.
Best regards,
Carlos Sierra

There is no code associated to this manuscript, which is stated in the data availability section. We have updated the Authors Contributions to

"P.Ciais designed and wrote the manuscript, with inputs by F. Chevallier for the inversion section and A. Bastos and J. Pongratz for the land use section. H. Yang additionally helped with references. All other contributors helped to improve the text in their field of expertise. P.Ciais and A. Bastos revised the manuscript and prepared the response to reviewers."